# Altering the redox status of *Chlamydia trachomatis* directly impacts its developmental cycle progression

Vandana Singh, Scot P Ouellette*

Department of Pathology, Microbiology, and Immunology, College of Medicine, University of Nebraska Medical Center, Omaha, United States

## eLife Assessment

In this **valuable** study, the authors propose a model wherein the bacterial redox state plays a crucial role in the differentiation of *Chlamydia trachomatis* into elementary and reticulate bodies. They provide **solid** evidence to argue that a highly oxidising environment favours the formation of elementary bodies while a reducing condition slows down development. Overall, the study **convincingly** demonstrates that Chlamydial redox states play a role in differentiation, an observation that may have implications for the study of other bacterial systems.

**\*For correspondence:**
scot.ouellette@unmc.edu

**Competing interest:** The authors declare that no competing interests exist.

**Abstract** *Chlamydia trachomatis* is an obligate intracellular bacterial pathogen with a unique developmental cycle. It differentiates between two functional and morphological forms: the elementary body (EB) and the reticulate body (RB). The signals that trigger differentiation from one form to the other are unknown. EBs and RBs have distinctive characteristics that distinguish them, including their size, infectivity, proteome, and transcriptome. Intriguingly, they also differ in their overall redox status as EBs are oxidized and RBs are reduced. We hypothesize that alterations in redox may serve as a trigger for secondary differentiation. To test this, we examined the function of the primary antioxidant enzyme alkyl hydroperoxide reductase subunit C (AhpC), a well-known member of the peroxiredoxins family, in chlamydial growth and development. Based on our hypothesis, we predicted that altering the expression of *ahpC* would modulate chlamydial redox status and trigger earlier or delayed secondary differentiation. Therefore, we created *ahpC* overexpression and knockdown strains. During *ahpC* knockdown, ROS levels were elevated, and the bacteria were sensitive to a broad set of peroxide stresses. Interestingly, we observed increased expression of EB-associated genes and concurrent higher production of EBs at an earlier time in the developmental cycle, indicating earlier secondary differentiation occurs under elevated oxidation conditions. In contrast, overexpression of AhpC created a resistant phenotype against oxidizing agents and delayed secondary differentiation. Together, these results indicate that redox potential is a critical factor in developmental cycle progression. For the first time, our study provides a mechanism of chlamydial secondary differentiation dependent on redox status.

## Introduction

All organisms that are exposed to oxygen are necessarily subjected to oxidative stress. Specifically, the process of metabolizing substrates in the presence of oxygen can generate reactive oxygen species (ROS), which are toxic at high enough concentrations. Thus, from bacteria to humans, systems have evolved to mitigate the accumulation of ROS. At the same time, host immune defense mechanisms have evolved to leverage ROS production as a means of limiting pathogen growth and survival.

Not surprisingly, pathogens have co-evolved to resist these defense mechanisms. For example, many pathogens possess a variety of antioxidant enzymes, such as catalases, glutathione peroxidases, and peroxiredoxins, that help them both subvert ROS-mediated immune system assaults and mitigate metabolic ROS byproducts (*Staerck et al., 2017*; *Wan et al., 2021*).

*Chlamydia* is an obligate intracellular bacterium that has significantly reduced its genome size and content in adapting to obligate host dependence. *Chlamydia trachomatis*, the leading cause of bacterial sexually transmitted diseases and preventable infectious blindness, lacks homologs to catalases or glutathione peroxidases but does possess a homolog of AhpC. AhpC is a 2-cys peroxiredoxin that is widely conserved in prokaryotes (*de Oliveira et al., 2021*). Peroxiredoxins scavenge hydrogen peroxide, peroxynitrite, and organic hydroperoxides (*Parsonage et al., 2008*; *Poole and Ellis, 1996*; *Seaver and Imlay, 2001*) and act as the primary scavenger in pathogens that lack both catalase and glutathione peroxidases (*Mastronicola et al., 2014*; *Richard et al., 2011*). Several studies from other bacterial systems have shown that AhpC has a significant role in ROS and RNI scavenging, virulence, and persistence (*Cosgrove et al., 2007*; *Kimura et al., 2012*; *Oh and Jeon, 2014*), and deletion of AhpC results in elevated levels of ROS within the bacterium (*Zhang et al., 2019*). Though *Chlamydia* is dependent on its host for most of its energy requirements, it has some metabolic activities, such as a partial TCA cycle and oxidative phosphorylation (*Gérard et al., 2002*; *Iliffe-Lee and McClarty, 1999*), which can be a possible source of intracellular ROS to which the bacteria must adapt. There is a paucity of knowledge on how *C. trachomatis* modulates oxidative stress. However, some studies explored the effects of redox changes on chlamydial growth. Boncompain et al. reported that infection of *C. trachomatis* induced the transient production of ROS by the host cell at a moderate level for the initial few hours of infection only (*Boncompain et al., 2010*). Another study also found a similar observation about ROS during infection, reporting that

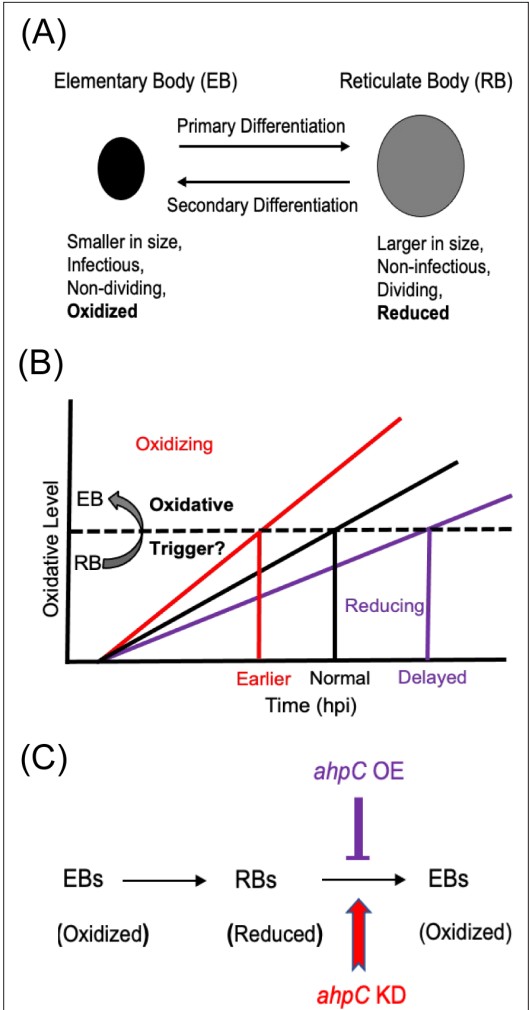

**Figure 1.** Alterations in the redox status of key proteins regulate and drive chlamydial differentiation. (**A**) Key characteristics of chlamydial developmental forms. (**B**) Hypothetical model for triggering secondary differentiation through oxidative stress (black angled line). Increasing oxidation of critical protein(s) may lead to earlier differentiation whereas maintaining a reducing environment may delay differentiation. (**C**) Schematic representation of the experimental model for triggering secondary differentiation through the altered activity of alkyl hydroperoxide reductase subunit C (AhpC). *ahpC* knockdown may lead to earlier differentiation, while overexpression of *ahpC* may delay differentiation.

*Chlamydia* requires host-derived ROS for its growth (*Abdul-Sater et al., 2010*). However, the key mechanisms *Chlamydia* employs to manage ROS-mediated stress have not been characterized.

*Chlamydia* undergoes a complex developmental cycle that comprises two distinct morphological forms, the EB and the RB (*Abdelrahman and Belland, 2005*). The EB is the smaller (~0.3 μm), infectious, and nondividing form capable of infecting susceptible host cells. Once internalized into a host-derived vacuole termed an inclusion, the EB differentiates into the non-infectious, larger (~1 μm), and replicating RB - this process is known as primary differentiation and represents the early phase of the developmental cycle (*Clifton et al., 2005*). RBs replicate within the inclusion in the midcycle phase

using an asymmetric, MreB-dependent polarized division process (*Abdelrahman et al., 2016*; *Lee et al., 2020*; *Ouellette et al., 2022*; *Ouellette et al., 2020*). In the late phase of the developmental cycle, RBs asynchronously condense into EBs, and this process is termed secondary differentiation. Despite these well-defined differences between chlamydial morphological forms, how *Chlamydia* mechanistically differentiates between functional forms remains unclear.

Intriguingly, a recent study evaluated the redox potential of *Chlamydia* and demonstrated that RBs are reduced whereas EBs are oxidized (*Wang et al., 2014*; *Figure 1A*). This is consistent with earlier studies that revealed differences in the crosslinking of outer membrane proteins and type III secretion-related proteins in chlamydial developmental forms (*Betts-Hampikian and Fields, 2011*; *Caldwell et al., 1981*; *Everett and Hatch, 1995*; *Wang et al., 2014*). Taken together, these observations indicate that, during the developmental cycle, the redox potential of *Chlamydia* is changing. However, whether redox changes in *Chlamydia* directly affect developmental cycle progression is not characterized.

Based on the difference between the redox status of EBs and RBs, we hypothesized that changing redox conditions is a critical factor in the process of differentiation from one form to the other. We named this the 'redox threshold hypothesis.' In this scenario, as soon as a given RB has crossed an oxidative threshold, the activity of critical proteins is modified to trigger differentiation to the EB (*Figure 1B*). We used chlamydial transformants designed to overexpress or reduce AhpC levels to explore the effects of altered redox potential on chlamydial growth and development to test this hypothesis (*Figure 1C*). This study establishes the role of AhpC as an antioxidant in *Chlamydia*, as demonstrated by its ability to counteract different peroxide stresses when overexpressed. Overexpression of AhpC had no negative effect on bacterial replication but delayed the differentiation of RBs to EBs. In contrast, under conditions of *ahpC* knockdown, the organism was highly sensitive to oxidizing conditions. Interestingly, this change in redox status caused earlier expression of EB-associated (late) genes and production of EBs, leading to a shift in developmental cycle progression. This earlier activation of gene expression related to secondary differentiation in *ahpC* knockdown was also observed when developmental cycle progression was blocked by penicillin treatment. Taken together these data provide mechanistic insight into chlamydial secondary differentiation and are the first to demonstrate redox-regulated differentiation in *Chlamydia*.

## Results

### Overexpression of *ahpC* favors RB replication over EB production

To study the effect of AhpC on chlamydial developmental cycle progression, we first generated an *ahpC* overexpression (OE) strain using a plasmid encoding an anhydrotetracycline (aTc) inducible *ahpC* (untagged) and transformed it into a *C. trachomatis* L2 strain lacking its endogenous plasmid (-pL2). The same plasmid vector backbone (i.e., encoding mCherry in place of *ahpC*) was used as an empty vector plasmid control (EV). HeLa cells were infected with these transformants, and, at 10 hpi, expression of the construct was induced or not. Lacking an antibody against AhpC, overexpression of *ahpC* was validated by reverse transcription-quantitative PCR (RT-qPCR). Here, an approximate 1-log increase in transcripts of *ahpC* was detected at 14 and 24 hpi in the induced strain in comparison to the uninduced strain (*Figure 2A*). Immunofluorescence analysis (IFA) was performed at 14 and 24 hpi to examine the organisms' morphology and overall inclusion growth. For IFA, individual bacteria were labeled with an anti-major outer membrane protein (MOMP) antibody, and DAPI was used to stain for DNA. IFA imaging revealed that there was no observable impact on the morphology of the EV control strain under the conditions tested. In contrast, we did note that overexpression of *ahpC* increased the overall inclusion area (*Figure 2B and C*). We next quantified the total number of bacteria (i.e., both RBs and EBs) by measuring genomic DNA and observed a significant increase in gDNA levels at 24 hpi in response to increased *ahpC* expression (*Figure 2D*). Although inclusions were larger in area and contained more total bacteria as assessed by gDNA levels when overexpressing *ahpC*, recoverable inclusion forming units (IFUs: a measure of infectious EBs) were significantly lower under these conditions at 24 hpi (*Figure 2E*). However, by 48 hpi, the difference between the induced and uninduced conditions, although still reduced, was not statistically significant. The average number of IFUs was $5.30 \times 10^7$ and $1.56 \times 10^9$ for the *ahpC* OE strain in the uninduced condition at 24 and 48 hpi, respectively. For the EV strain, the average number of IFUs was $7.07 \times 10^6$ and $1.90 \times 10^8$ in the uninduced

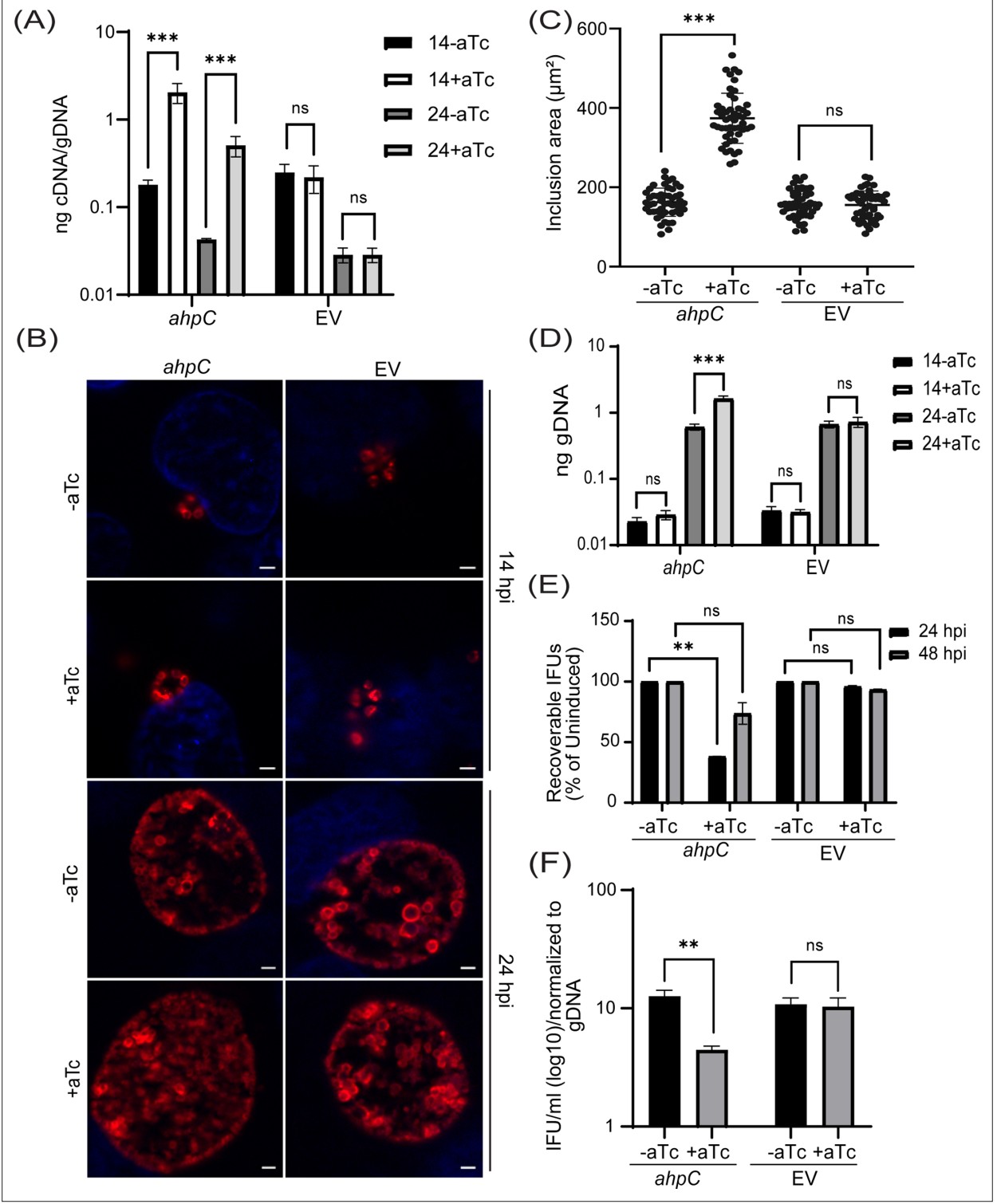

**Figure 2.** Overexpression of alkyl hydroperoxide reductase subunit C (*ahpC*) affects chlamydial growth and differentiation. (**A**) Transcriptional analysis of *ahpC* in *ahpC* overexpression (*ahpC*) and empty vector (EV) control using RT-qPCR following induction at 10 hpi with 1 nM aTc. RNA and genomic DNA (gDNA) were harvested at 14 and 24 hpi and processed as mentioned in the materials and methods. Data are presented as a ratio of cDNA to gDNA plotted on a log scale. ***p<0.0001 vs uninduced sample by using two-way ANOVA and Tukey's HSD was applied as a post hoc test. Data represent three biological replicates. (**B**) Immunofluorescence assay (IFA) of *ahpC* and EV at 14 and 24 hpi. Construct expression was induced or not at 10 hpi with 1 nM aTc, and samples were fixed with methanol at 14 and 24 hpi and then stained for major outer membrane protein (MOMP - red) and DAPI (blue) to label DNA. Scale bars = 2 μm. Images were captured using a Zeiss Axio Imager Z.2 with Apotome2 at 100 x magnification. Representative images of

*Figure 2 continued on next page*

*Figure 2 continued*

three biological replicates are shown. (**C**) Impact of *ahpC* overexpression on inclusion area. Inclusion area of *ahpC* overexpression and EV strains was measured using ImageJ. Experimental conditions were the same as mentioned in section (**B**). The area of 50 inclusions was measured per condition for each sample. ***p<0.001 vs uninduced sample by using ordinary one-way ANOVA and Tukey's HSD was applied as a post hoc test. Data were collected from three biological replicates. (**D**) Quantification of genomic DNA (gDNA) determined by qPCR in *ahpC* overexpression and empty vector control. Construct expression was induced or not at 10 hpi with 1 nM aTc, gDNA was harvested at 14 and 24 hpi, and ng gDNA was plotted on a log scale. ***p<0.0001 vs uninduced sample by using two-way ANOVA and Tukey's HSD was applied as a post hoc test. Data represent three biological replicates. (**E**) IFU assay of *ahpC* overexpression and empty vector control. Expression of the construct was induced or not at 10 hpi, and samples were harvested at 24 or 48 hpi for reinfection and enumeration. IFUs were calculated as the percentage of uninduced samples. **p<0.001 vs uninduced sample by using multiple paired t-test. Data represent three biological replicates. (**F**) Ratio of log10 IFUs and log10 gDNA. IFU/ml from (**E**) was normalized with gDNA from (**D**). **p<0.001 vs uninduced sample by using multiple unpaired t-test. Data represent three biological replicates.

The online version of this article includes the following source data for figure 2:

**Source data 1.** RT-qPCR (cDNA and gDNA), quantification of inclusion size, gDNA, and IFU data of *ahp*C OE and EV strains.

condition at 24 and 48 hpi, respectively. The inclusion forming unit analysis only measures viable EBs from a population, suggesting that AhpC overexpression skews the ratio of RBs to EBs. To test this, we calculated the ratio of IFUs to gDNA at 24 hpi, which revealed that the relative number of RBs is higher (and EBs lower) as a result of overexpression of *ahpC* (*Figure 2F*). Overall, these data support the hypothesis that overexpressing *ahpC* delays production of infectious EBs.

## Overexpression of *ahpC* confers resistance to peroxides in *Chlamydia*

AhpC is an important peroxiredoxin involved in oxidative damage defense. Before determining whether increased *ahpC* expression impacts chlamydial sensitivity to oxidizing agents, we first sought to determine the response of *C. trachomatis* to inorganic or organic hydroperoxides such as hydrogen peroxide ($H_2O_2$), cumene hydroperoxide (CHP), and tert-butyl hydroperoxide (TBHP) as well as peroxynitrite (PN). To do this, we evaluated the sensitivity of HeLa cells, infected or not with the EV control strain, to different concentrations of oxidizing agents using a viability assay. Both uninfected and EV-infected HeLa cells tolerated up to 1 mM oxidizing agents for 30 min, retaining more than 90% viability (*Figure 3—figure supplement 1*). Next, different concentrations (lower than 1 mM) of these oxidants were tested on wild-type (WT) *C. trachomatis* L2/434/Bu (*Ctr* L2). HeLa cells were infected with *Ctr* L2 and, at 16 hpi, were exposed to different concentrations of exogenous oxidizing agents for 30 min only, before washing out the oxidizing agents and replacing the media. Samples for IFU and IFA assays were then collected at 24 hpi to assess the effects of the oxidizing agents on WT *Ctr* L2 IFU recovery. 62.5 µM concentration of all three inorganic and organic peroxides had no appreciable effect on IFUs, and inclusion size and morphology also remained unaffected at this sublethal concentration. The concentrations of inorganic and organic peroxides that resulted in a decrease in IFUs to ~50% of the untreated culture were 500 µM ($H_2O_2$) and 250 µM (CHP and TBHP), respectively, with a concurrent decrease in inclusion size. 1 mM $H_2O_2$, CHP, and TBHP caused >90% reduction in IFUs and small inclusions. PN at 1 mM concentration was less effective than other oxidizing agents and showed only a ~30% decline in IFUs compared to the untreated culture (*Figure 3—figure supplement 2*).

To further examine the antioxidant functions of AhpC, we next exploited our *ahpC* overexpression strain to explore its capacity to protect chlamydiae from oxidizing agents. Both *ahpC* OE and EV strains were used to infect HeLa cells and, at 10 hpi, expression of the constructs was induced or not with 1 nM aTc. At 16 hpi, infected cells were exposed to various concentrations of exogenous oxidizing agents for 30 min. At 24 hpi, IFU and IFA samples were collected to measure infectious EB production and assess chlamydial morphology, respectively. As shown in *Figure 3A and B*, and Supplement 3, the *ahpC* overexpression strain showed increased resistance to all oxidants tested as the mean IFUs and the size of inclusions were greater than that of the uninduced but treated control. There was no change in bacterial growth and morphology in the case of 62.5 µM concentrations of oxidizing agents, which was anticipated since we determined this was a sublethal concentration. The resistant phenotype was more evident at the higher concentrations of $H_2O_2$, CHP, and TBHP. Overexpression of *ahpC* also contributed to higher resistance against PN. Conversely, the empty vector control strain behaved as expected (i.e., like WT) in the presence of oxidizing agents, with similar results observed in the presence or absence of aTc (*Figure 3C, D*, *Figure 3—figure supplement 3*). Collectively, these data

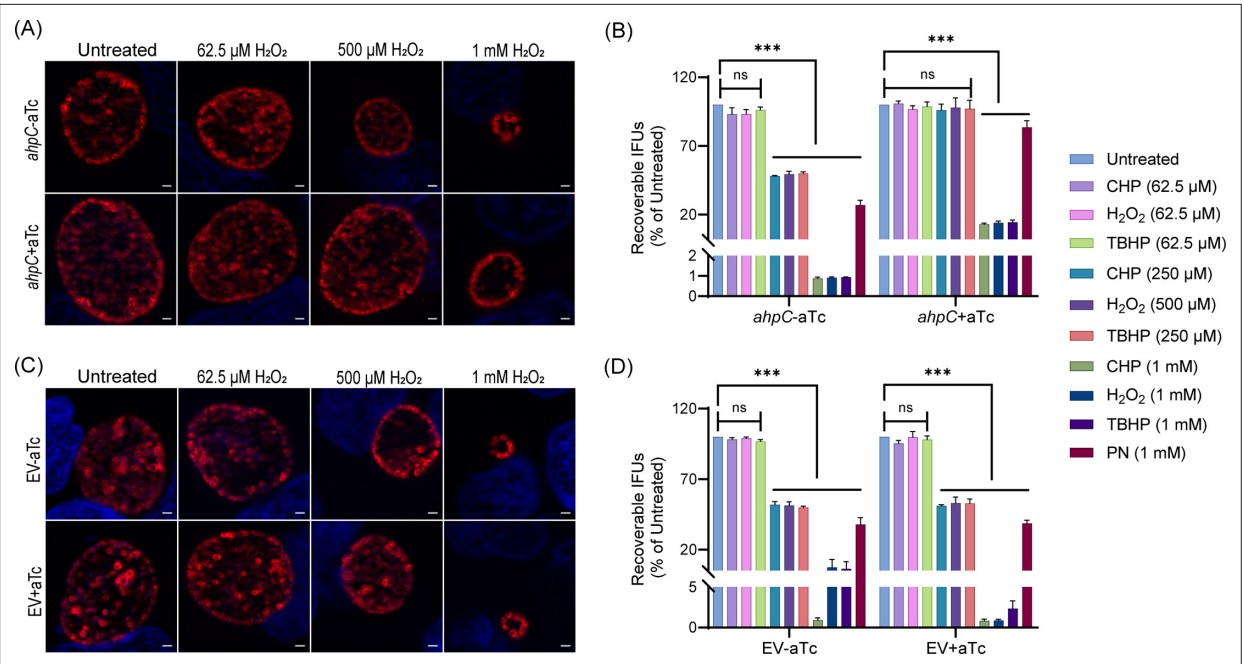

**Figure 3.** Higher expression of alkyl hydroperoxide reductase subunit C (*ahpC*) provides resistance to peroxides in *Chlamydia*. Immunofluorescence analysis (IFA) of *ahpC* (**A**) or empty vector (EV) (**C**) exposed to oxidizing agents. Construct expression was induced or not at 10 hpi with 1 nM aTc, and samples were treated with three different concentrations of hydrogen peroxide (H₂O₂) at 16 hpi for 30 min, then fixed with methanol at 24 hpi, stained and imaged as described in the legend of *Figure 2B*. Representative images from three biological replicates are shown. Scale bars = 2 µm. IFU analysis of *ahpC* (**B**) or EV (**D**) following treatment with oxidizing agents, CHP-Cumene hydroperoxide, H₂O₂-Hydrogen peroxide, TBHP-Tert-butyl hydroperoxide, and PN-Peroxynitrite. Samples were processed as described for (**A**) and (**C**), and IFUs were harvested at 24 hpi. IFUs of treated samples were compared with respective untreated controls. ***p<0.0001 vs untreated sample by using two-way ANOVA and Tukey's HSD was applied as a post hoc test. Data represent three biological replicates.

The online version of this article includes the following source data and figure supplement(s) for figure 3:

**Source data 1.** IFU data of *ahp*C OE and EV against CHP, hydrogen peroxide, TBHP, and PN.

**Figure supplement 1.** Viability assay of uninfected or infected HeLa cells treated with oxidizing agents hydrogen peroxide (H₂O₂) (**A**), cumene hydroperoxide (CHP) (**B**), tert-butyl hydroperoxide (TBHP) (**C**), and peroxynitrite (PN) (**D**).

**Figure supplement 1—source data 1.** Viability data of uninfected or infected HeLa cells in presence of hydrogen peroxide, CHP, TBHP, and PN.

**Figure supplement 2.** Response of *C.trachomatis* L2 against oxidizing agents.

**Figure supplement 2—source data 1.** IFU data of *Ctr* L2 against CHP, hydrogen peroxide, TBHP, and PN.

**Figure supplement 3.** Overexpression of alkyl hydroperoxide reductase subunit C (*ahpC*) provides resistance to peroxides in *Chlamydia*.

demonstrate that the chlamydial AhpC possesses antioxidant activity, as expected, and that increased *ahpC* expression is protective for *Chlamydia* in the presence of increased oxidative stress.

## Knockdown of *ahpC* negatively impacts chlamydial growth

To further define the function(s) of AhpC in the chlamydial developmental cycle, we used a novel dCas12-based CRISPR interference (CRISPRi) strategy adapted for *Chlamydia* by our lab (*Ouellette et al., 2021*). We generated an *ahpC* knockdown strain harboring the pBOMBL12CRia CRISPRi plasmid with a crRNA targeting the *ahpC* 5' intergenic region and overlapping the ATG start site (plasmid designated as pL12CRia(*ahpC*)). We used a strain carrying a pL12CRia plasmid with a crRNA with no homology to any chlamydial sequence (i.e., non-targeting [NT]) to serve as a negative control. To confirm the knockdown of *ahpC*, RT-qPCR was employed. Here, we infected HeLa cells with the *ahpC* knockdown (*ahpC* KD) or NT strains and induced dCas12 expression or not at 10 hpi using 1 nM aTc. Nucleic acid samples were harvested at 14 hpi and 24 hpi. RT-qPCR analysis of the *ahpC* KD revealed approximately 90% reduction of *ahpC* transcripts compared to the uninduced control, thus confirming the knockdown of *ahpC* (*Figure 4A*). In the case of the NT control, there was no effect on *ahpC* transcripts. IFA of these strains revealed noticeably smaller inclusions after blocking

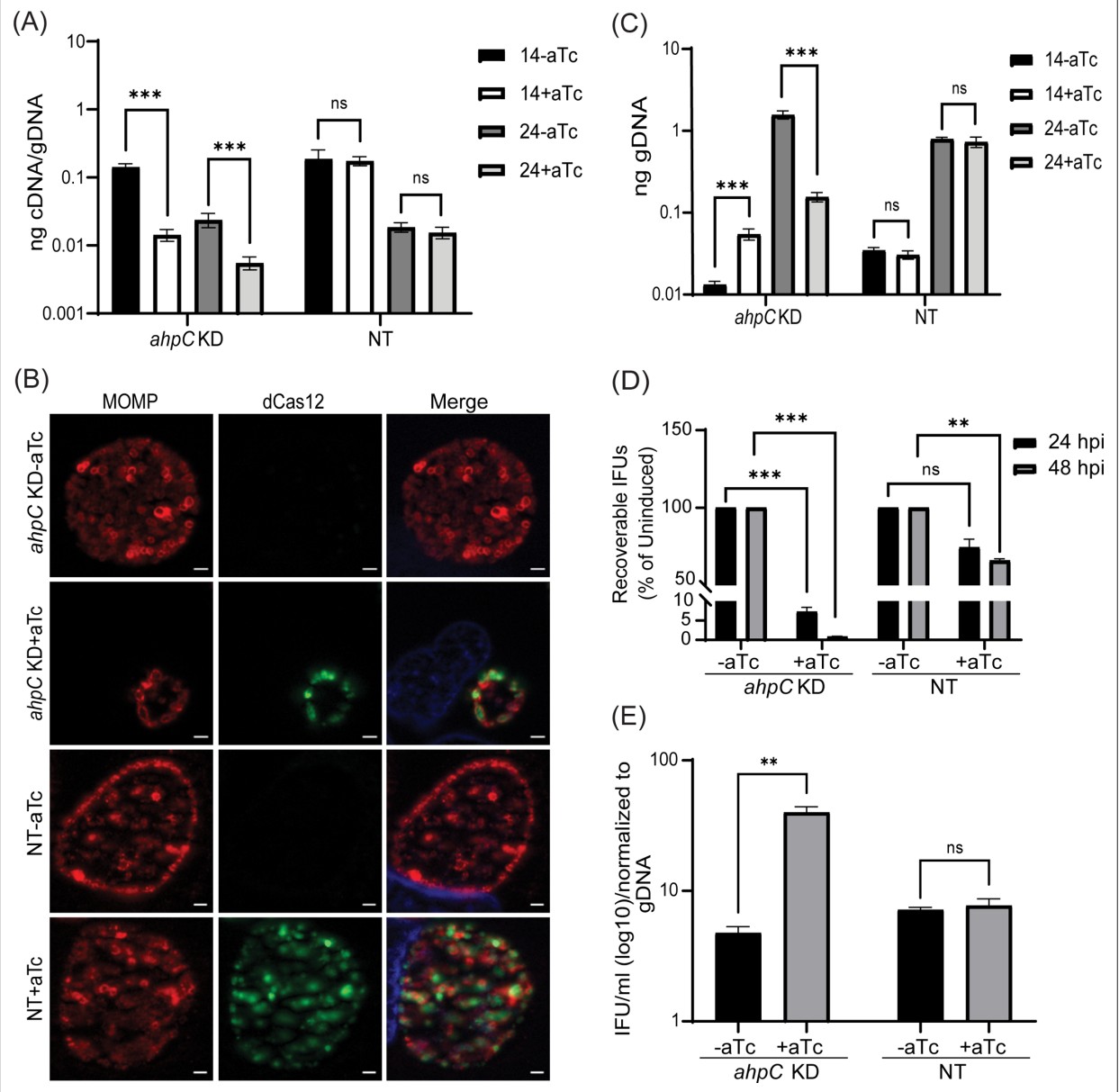

**Figure 4.** Reduced levels of alkyl hydroperoxide reductase subunit C (AhpC) negatively impact chlamydial growth. (**A**) Transcriptional analysis of *ahpC* in knockdown (*ahpC* KD) and non-target (NT) control using RT-qPCR following induction at 10 hpi with 1 nM aTc. RNA and gDNA were harvested at 14 and 24 hpi. Quantified cDNA was normalized to gDNA, and values were plotted on a log scale. ***$p<0.0001$ vs uninduced sample by using two-way ANOVA and Tukey's HSD was applied as a post hoc test. Data represent three biological replicates. (**B**) Immunofluorescence analysis (IFA) was performed to assess inclusion size and morphology using the same induction conditions as in section (**A**). At 24 hpi, cells were fixed with methanol and stained using primary antibodies to major outer membrane protein (MOMP), Cpf1 (dCas12), and DAPI. All images were acquired on Zeiss Axio Imager Z.2 with Apotome2 at 100 x magnification. Scale bars = 2 μm. Representative images of three biological replicates are shown. (**C**) Quantification of genomic DNA (gDNA) determined by qPCR in *ahpC* KD and NT strains. dCas12 expression was induced or not at 10 hpi, and gDNA was harvested at 14 and 24 hpi and plotted on a log scale. ***$p<0.0001$ vs uninduced sample by using two-way ANOVA and Tukey's HSD was applied as a post hoc test. Data represent three biological replicates. (**D**) IFU titers following induction at 10 hpi with 1 nM aTc. IFUs were counted from 24 and 48 hpi samples and calculated as a percentage of uninduced samples. ***$p<0.0001$, **$p<0.001$ vs uninduced sample by using multiple paired t-test. Data represent three biological replicates. (**E**) Ratio of log10 IFUs by log10 gDNA. IFU/ml from (**D**) was normalized with gDNA from (**C**). **$p<0.001$ vs uninduced sample by using multiple unpaired t-test. Data represent three biological replicates.

The online version of this article includes the following source data for figure 4:

**Source data 1.** RT-qPCR (cDNA and gDNA), gDNA, and IFU data of *ahpC* KD and NT strains.

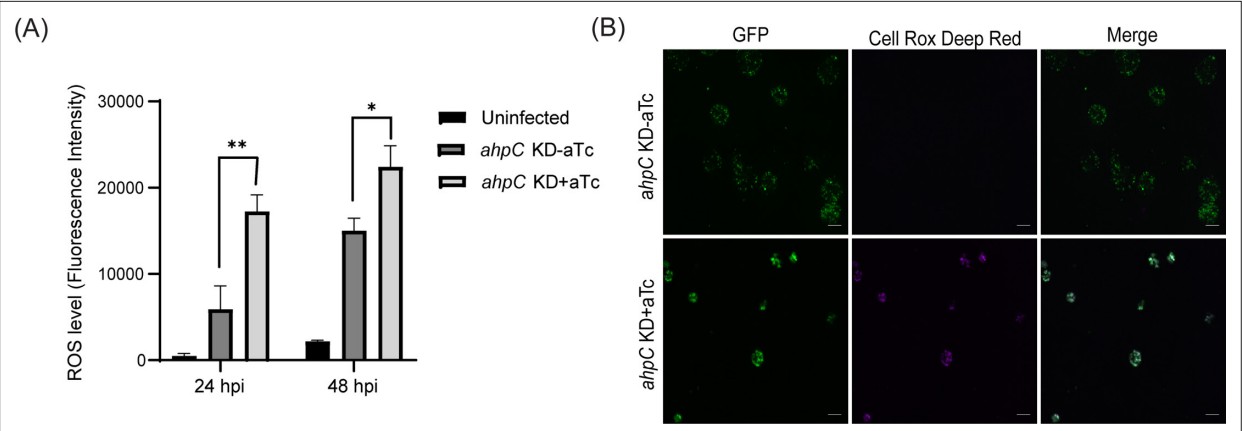

**Figure 5.** Intracellular reactive oxygen species (ROS) levels were measured to investigate the function of alkyl hydroperoxide reductase subunit C (AhpC) in reducing ROS. (**A**) HeLa cells were infected or not with *ahpC* knockdown, and the construct was induced or not at 10 hpi with 1 nM aTc. At 24 or 48 hpi, samples were washed with DPBS and incubated with CellROX Deep red dye for 30 min in the dark. ROS levels were measured at wavelengths of 640 nm (excitation) and 665 nm (emission). **p<0.001, *p<0.01 vs uninduced sample by using two-way ANOVA, and Tukey's HSD was applied as a post hoc test. Data represent three biological replicates. (**B**) Microscopy images were acquired using live cells on Zeiss Axio Imager Z.2 with Apotome2 at 100 x magnification. Scale bars = 10 μm. Representative images of three biological replicates are shown.

The online version of this article includes the following source data and figure supplement(s) for figure 5:

**Source data 1.** Values of ROS measurement in uninfected HeLa cells and *ahpC* KD strain in the uninduced and induced conditions.

**Figure supplement 1.** Intracellular reactive oxygen species (ROS) levels in alkyl hydroperoxide reductase subunit C (*ahpC*) knockdown at 40 hpi.

expression of *ahpC* as compared to the uninduced sample or the NT conditions (*Figure 4B*). We next measured total bacterial counts using genomic DNA. Surprisingly, even though inclusions were smaller during *ahpC* knockdown, we observed higher gDNA levels at 14 hpi compared to the uninduced control, followed by only a small increase from 14 to 24 hpi (*Figure 4C*). In contrast, the uninduced strain showed a logarithmic increase in gDNA levels during this timeframe, similar to the NT strain (*Figure 4C*). To further explore the effect of reduced *ahpC* transcripts in *Chlamydia*, IFU assays were performed. IFU analysis exhibited severely reduced progeny (>90%) during *ahpC* knockdown conditions compared to its respective uninduced control at both time points assessed (24 and 48 hpi) (*Figure 4D*). In contrast, the NT strain showed less than a 50% reduction after inducing dCas12 expression at these timepoints consistent with prior observations that dCas12 expression slightly delays developmental cycle progression (*Hatch and Ouellette, 2023*; *Reuter et al., 2023*). The ratio of IFUs to gDNA at 24 hpi is significantly higher for the *ahpC* KD, suggesting a lower number of RBs in proportion to EBs (*Figure 4E*). Taken together, these analyses indicate that reducing *ahpC* levels and/ or activity severely reduced chlamydial growth at 24 and 48 hpi.

## Knockdown of *ahpC* increases bacterial ROS levels

To investigate whether *ahpC* knockdown resulted in increased ROS levels in the bacteria (*Zhang et al., 2019*), the intracellular ROS levels were measured in infected cells in *ahpC* knockdown conditions and compared to the uninduced control condition and uninfected cells. We used the cell-permeable, fluorogenic dye CellROX Deep Red to measure ROS levels. This dye remains non-fluorescent in a reduced state and exhibits bright fluorescence once oxidized by ROS. ROS generation was measured in uninfected and *ahpC* KD-infected HeLa cells at 24 and 48 hpi. As shown in *Figure 5A*, *ahpC* knockdown resulted in significantly higher ROS than the uninfected or infected but uninduced samples at both 24 hpi and 48 hpi time points. We also performed live-cell microscopy for visualization of ROS generation in uninfected and *ahpC* KD-infected HeLa cells at 24 and 40 hpi. These microscopy images revealed that *ahpC* KD resulted in higher ROS in comparison to the uninduced and uninfected cells (*Figure 5B*, *Figure 5—figure supplement 1*). Importantly, the signal was associated with chlamydial inclusions and not the host cell, thus indicating that the measurements from *Figure 5A* were from the bacteria. These results together indicate that AhpC knockdown results in increased ROS levels in *C. trachomatis*.

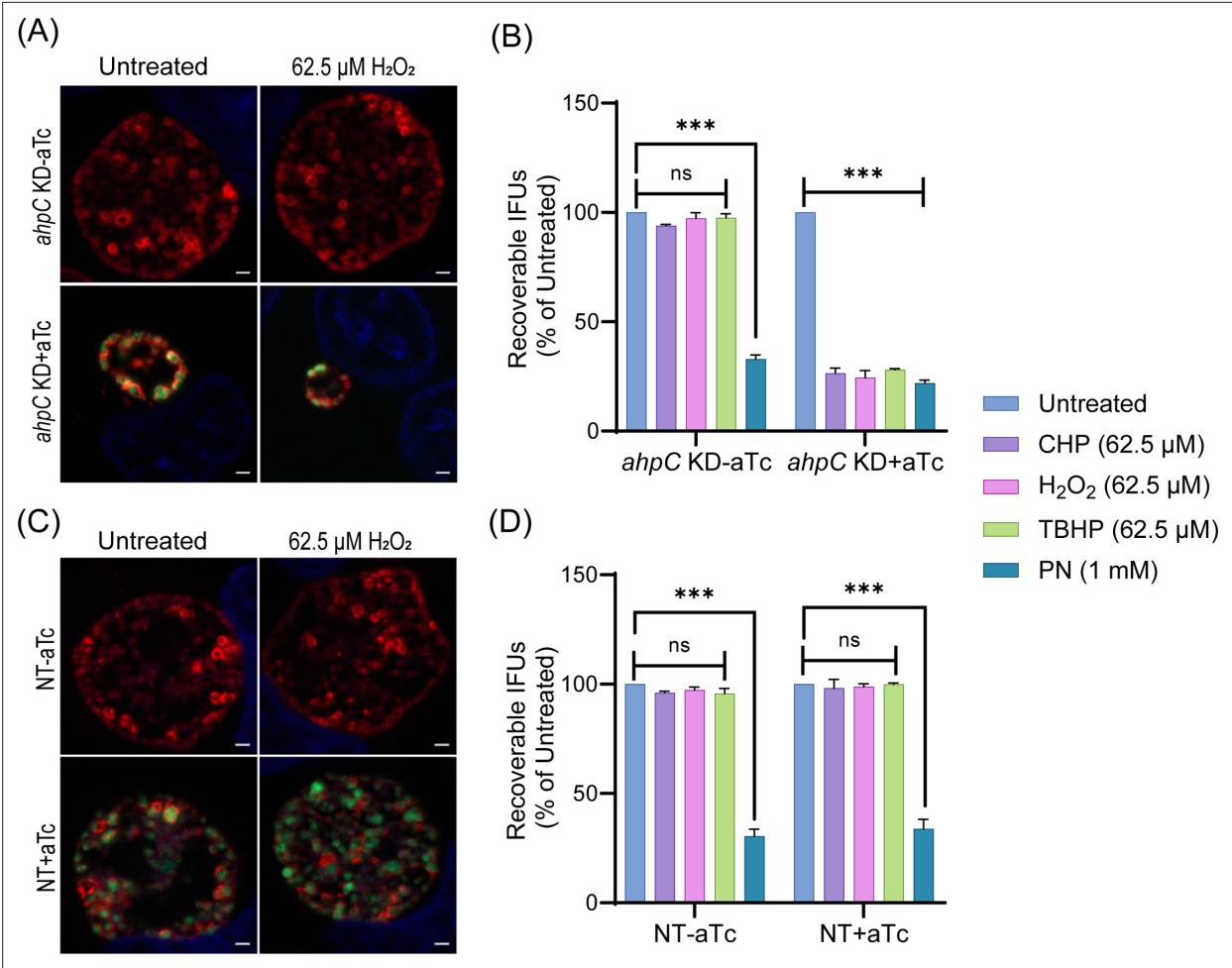

**Figure 6.** *Chlamydia* is hypersensitive to oxidizing agents in alkyl hydroperoxide reductase subunit C (*ahpC*) knockdown condition. Immunofluorescence analysis (IFA) of *ahpC* KD (**A**) or NT (**C**) treated with 62.5 μM H₂O₂. dCas12 expression was induced or not at 10 hpi with 1 nM aTc, treated or not with H₂O₂ at 16 hpi for 30 min, and allowed to grow until 24 hpi. Coverslips were fixed with methanol at 24 hpi and stained major outer membrane protein (MOMP), Cpf1 (dCas12), and DAPI. Scale bars = 2 μm. Images were captured using a Zeiss Axio Imager Z.2 with Apotome2 at 100 x magnification. Representative images from three biological replicates are shown. IFU analysis of *ahpC* KD (**B**) or NT (**D**) following treatment with oxidizing agents, CHP-Cumene hydroperoxide, H₂O₂-Hydrogen peroxide, TBHP-Tert-butyl hydroperoxide, or PN-Peroxynitrite. dCas12 expression was induced or not, and samples were treated or not as mentioned in the legend of *Figure 3B*. IFUs of treated samples were calculated as a percentage of respective untreated samples. ***$p<0.0001$ vs untreated sample by using two-way ANOVA and Tukey's HSD was applied as post hoc test. Data represent three biological replicates.

The online version of this article includes the following source data and figure supplement(s) for figure 6:

**Source data 1.** IFU values after CHP, hydrogen peroxide, TBHP, and PN treatment in the *ahp*C KD and NT strains.

**Figure supplement 1.** *Chlamydia* is hypersensitive to oxidizing agents as a result of reduced levels of alkyl hydroperoxide reductase subunit C (*ahpC*).

## Knockdown of *ahpC* sensitizes *Chlamydia* to oxidizing agents

As overexpression of *ahpC* resulted in increased resistance to oxidizing agents, we next explored whether knockdown of *ahpC* resulted in increased sensitivity to such agents. As described above, *ahpC* KD and NT strains were used to infect HeLa cells and, at 10 hpi, expression of dCas12 was induced or not with 1 nM aTc. Considering that the *ahpC* KD already demonstrated reduced IFUs and inclusion sizes at 24 hpi, only sublethal concentrations (62.5 μM) of exogenous oxidizing agents were applied. At 24 hpi, IFA and IFU analyses were performed to quantify the effect of the oxidants on chlamydial growth in the absence of *ahpC* activity. Under conditions of *ahpC* knockdown, even sublethal concentrations (62.5 μM) of H₂O₂, CHP, or TBHP further reduced the inclusion size, and the IFU data revealed a decrease from >90% to <30% in comparison to the untreated control (***Figure 6A, B***, ***Figure 6—figure supplement 1***). Reduced expression of *ahpC* also affected the survival of *Chlamydia*

against peroxynitrite. Notably, there was no significant change in the NT control strain in the uninduced and induced samples in response to oxidizing agents as assessed by IFU and IFA (*Figure 6C, D*, *Figure 6—figure supplement 1*). These data further support that the chlamydial AhpC is a critical antioxidant enzyme in these bacteria.

## Complementation restores the growth and resistance to low levels of peroxide stress of the *ahpC* knockdown strain

To validate that the impaired growth, altered inclusion morphology, and enhanced sensitivity to peroxides were due to the decreased level/activity of AhpC during knockdown, we generated a complemented strain to restore *ahpC* expression during knockdown. For the construction of the *ahpC* complementing plasmid, the *ahpC* gene was cloned and transcriptionally fused 3' to the dCas12 in the pL12CRia(*ahpC*) knockdown plasmid. Here, the complementing *ahpC* allele is also under the control of the aTc-induced P*tet* promoter and is co-expressed with the aTc-inducible dCas12. Consequently, the *ahpC* knockdown effect is ablated, and the observed phenotypes should be restored. After inducing *dCas12-ahpC* expression with aTc, the resultant strain was verified by RT-qPCR. Increased transcripts for *ahpC* were quantified under these conditions, indicating successful complementation of the knockdown effect (*Figure 7A*). Of note, the *ahpC* transcript levels remained elevated in comparison to the uninduced control at the 24 hpi time point. After confirming this strain, we measured genomic DNA to quantify total bacteria and performed IFA and IFU assays to examine if complementation restored the phenotypes observed during *ahpC* knockdown. These assays revealed normal inclusion morphology, gDNA levels, and EB progeny production in the complemented strain, indicating successful complementation of the knockdown phenotype (*Figure 7B, C and D*). Of note, a C-terminal 6xHis tagged AhpC was not capable of complementing the knockdown phenotype (data not shown), indicating a requirement for a free C-terminus in the function of AhpC in *Chlamydia*. Previous studies in other bacteria have revealed that the C-terminal residues in AhpC play a crucial role in the structural stability and enzymatic activity of AhpC (*Dip et al., 2014*; *Feng et al., 2020*; *Wan et al., 2021*).

We next assessed whether complementation of the knockdown phenotype could also restore the resistance to low levels of peroxide stress. As in previous experiments, a sublethal concentration of oxidizing agents was added for 30' at 16 hpi after having induced *dCas12-ahpC* expression at 10 hpi. Consistent with the growth parameters (*Figure 7B–D*), the complemented strain showed wild-type responses in these conditions (*Figure 7E, F*, *Figure 6—figure supplement 1*). These data indicate that the enhanced susceptibility of the *ahpC* knockdown strain to oxidizing agents was due to the reduced levels of *ahpC* and not an indirect effect of knockdown.

We predicted that the growth defects resulting from higher production of ROS during *ahpC* knockdown could be rescued by treating the *ahpC* KD with ROS scavengers. To test this prediction, we utilized two characterized ROS scavengers, DMTU (*N,N'*-dimethylthiourea) and α-tocopherol, which scavenge $H_2O_2$ and peroxyl radical, respectively (*Holländer-Czytko et al., 2005*; *Kiffin et al., 2006*; *Walch et al., 2015*). Firstly, the effect of ROS scavengers on uninfected and infected HeLa cells was investigated using a viability assay. This assay revealed that 10 mM DMTU and 100 µM α-tocopherol had no adverse effects on uninfected or EV-infected HeLa cells (data not shown). These same concentrations were tested on WT *Ctr* L2 infected HeLa cells in untreated and 500 µM $H_2O_2$ treated conditions to test the potency of these scavengers to rescue growth defects associated with this concentration of oxidizing agent. Scavengers were added at 9.5 hpi and washed away at 16 hpi - the time of addition of $H_2O_2$ in the respective samples. At 16.5 hpi, after 3 wash steps, scavengers were added again with fresh media, and, at 24 hpi, bacterial growth and morphology were assessed using IFU and IFA assays. Neither scavenger had a significant impact on chlamydial morphology or infectious progeny. In the case of peroxide (500 µM $H_2O_2$) treated samples, scavengers restored IFUs from ~50% to ~100%, and inclusion size was also recovered (*Figure 7—figure supplement 1*).

Next, the effect of the ROS scavengers under *ahpC* knockdown conditions was assessed. In this experiment, scavengers were added at 9.5 hpi to provide the protective effect of scavengers before reducing the activity of *ahpC*. At 10 hpi, knockdown was induced or not with 1 nM aTc, and, at 14 hpi, scavengers were added again. At 24 hpi, IFU and IFA samples were collected to examine bacterial growth and morphology to assess the effect of ROS scavenging on the *ahpC* knockdown phenotype. As shown in *Figure 7G and H*, both ROS scavengers had a positive impact on restoring the growth of the *ahpC* knockdown strain; inclusions were larger, and IFUs were increased from <10% to >90% in

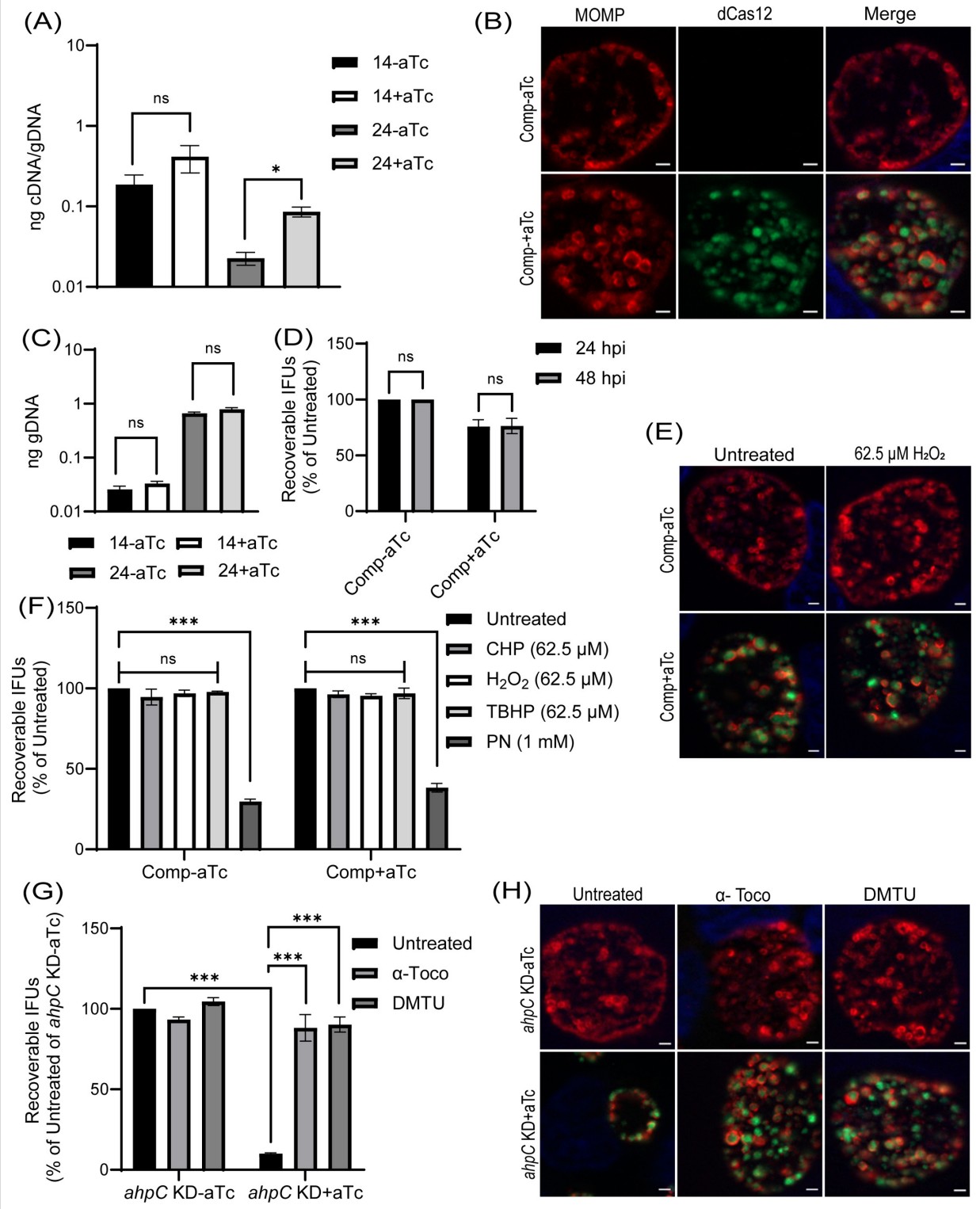

**Figure 7.** Complementation of the phenotypes observed in the alkyl hydroperoxide reductase subunit C (*ahpC*) knockdown. (**A**) Confirmation of complementation (comp) of *ahpC* knockdown by RT-qPCR. Samples were processed and quantified as mentioned previously in the legend of *Figure 4A*. Values were plotted on a log scale. *p<0.01 vs uninduced sample using ordinary one-way ANOVA and Tukey HSD was applied as post hoc test. Data represent three biological replicates. (**B**) Immunofluorescence analysis (IFA) of comp strain was performed at 24 hpi, and staining and imaging was performed as mentioned in the legend of *Figure 4B*. Scale bars = 2 µm. Representative images from three biological replicates are shown. (**C**) Genomic DNA quantitation was performed by qPCR. Construct expression was induced or not at 10 hpi, and gDNA was harvested at 14 and 24 hpi

*Figure 7 continued on next page*

*Figure 7 continued*

and plotted on a log scale. Statistical analysis was calculated using ordinary one-way ANOVA and Tukey's HSD was applied as a post hoc test. Data represent three biological replicates. (**D**) IFU analysis of comp strain. Statistical analysis was calculated using multiple paired t-test. Data represent three biological replicates. (**E**) IFA of comp strain following treatment with 62.5 μM $H_2O_2$. Samples were treated, stained, and images acquired as mentioned in the legend of *Figure 6A*. Scale bars = 2 μm. Representative images from three biological replicates are shown. (**F**) IFU analysis of comp strain following treatment with oxidizing agents. Experiments were performed as mentioned in the legend of *Figure 6B*. IFUs were calculated as a percentage of respective untreated samples. ***$p<0.0001$ vs untreated sample by using two-way ANOVA and Tukey's HSD was applied as a post hoc test. Data represent three biological replicates. (**G**) *ahpC* knockdown growth defect rescued by ROS scavengers. IFU analysis of *ahpC* knockdown treated with or without scavengers, α-Tocopherol (100 μM) and DMTU (10 mM), as mentioned in materials and methods. IFUs were calculated as a percentage of the untreated, uninduced sample. ***$p<0.0001$ vs untreated, uninduced, or induced sample by using two-way ANOVA and Tukey's HSD was applied as a post hoc test. Data represent three biological replicates. (**H**) IFA of *ahpC* knockdown treated with or without scavengers. Experimental conditions were similar as in section (**G**). Staining and imaging were performed as mentioned in *Figure 6A*. Representative images from three biological replicates are shown. Scale bars = 2 μm.

The online version of this article includes the following source data and figure supplement(s) for figure 7:

**Source data 1.** RT-qPCR (cDNA and gDNA), gDNA, and IFU after hydrogen peroxide stress in the complemented strain and IFU data in presence of scavengers in the *ahp*C KD strain.

**Figure supplement 1.** Rescue of oxidative stress phenotype by reactive oxygen species (ROS) scavengers in *C. trachomatis* L2.

**Figure supplement 1—source data 1.** IFU data of Ctr L2 in response to scavengers addition after treatment with hydrogen peroxide.

the presence of scavengers in the induced samples. These data provide compelling evidence that the adverse effects of *ahpC* knockdown are due to increased ROS accumulation in this strain.

## Chlamydial developmental cycle progression is altered by *ahpC* knockdown/overexpression

Whereas the overexpression of AhpC delayed overall developmental progression and was consistent with our hypothesis, the results with the *ahpC* knockdown strain appeared to refute our hypothesis given the apparent decrease in IFUs and smaller inclusion sizes after knockdown. To investigate the effects of changes in redox potential on chlamydial developmental cycle progression, we performed a transcriptional analysis of well-characterized late-cycle genes associated with secondary differentiation (*hctA*, *hctB*, *glgA*, *tsp*, and *omcB*) (*Barry et al., 1993*; *Belland et al., 2003*; *Brickman and Hackstadt, 1993*; *Gehre et al., 2016*; *Newhall, 1987*; *Swoboda et al., 2023*). In the uninduced samples, transcript levels of all the tested late genes were higher at 24 hpi compared to 14 hpi, indicating their normal expression during the late stage of the developmental cycle. In comparison to the uninduced control, under conditions of *ahpC* knockdown, significantly higher expression of *hctA*, *hctB*, *omcB*, *tsp*, and *glgA* was observed at the mid-developmental cycle timepoint of 14 hpi (*Figure 8A*). At 24 hpi, expression of these genes was comparable between the uninduced and induced conditions in the *ahpC* knockdown strain. We subsequently performed RNA sequencing from the uninduced and induced *ahpC* knockdown strain at 14 hpi. Experimental conditions were the same as described for the RT-qPCR study, and samples were processed as mentioned in Materials and Methods. RNA-seq results were statistically analyzed by the UNMC Bioinformatics Core facility (*Supplementary file 1*). We further categorized these genes based on significant differences ($p<0.05$) and fold-change ($\geq 1.5$). Using these stringent parameters, we identified 161 up-regulated genes and 145 down-regulated genes in the *ahpC* knockdown conditions (*Supplementary file 1*). Here, we observed a global increase in late gene transcripts at this mid-developmental cycle timepoint (*Figure 8B* and *Table 1*), indicating that increased oxidation results in earlier increases in late gene transcripts. In contrast, the complementation strain showed no increase in the expression of these tested late genes at 14 hpi, and *tsp* and *glgA* transcripts were reduced (*Figure 8—figure supplement 1A*). Consistent with our model, the overexpression of *ahpC* resulted in significantly lower expression of *hctA*, *hctB*, *omcB*, *tsp*, and *glgA* at 14 and 24 hpi (*Figure 8—figure supplement 1B*), suggesting a delayed transition to EBs.

Given the increase in late gene expression during *ahpC* knockdown at an earlier time point in the developmental cycle than normal, we reasoned that the increase in oxidizing conditions might prematurely trigger secondary differentiation and EB production. However, complicating such an analysis is that fewer RBs are present to convert to EBs, which would result in overall lower IFU yields as we had measured at 24 and 48 hpi (*Figure 4D*). Nonetheless, to assess EB production more rigorously at earlier time points in the developmental cycle, we performed a growth curve analysis to precisely

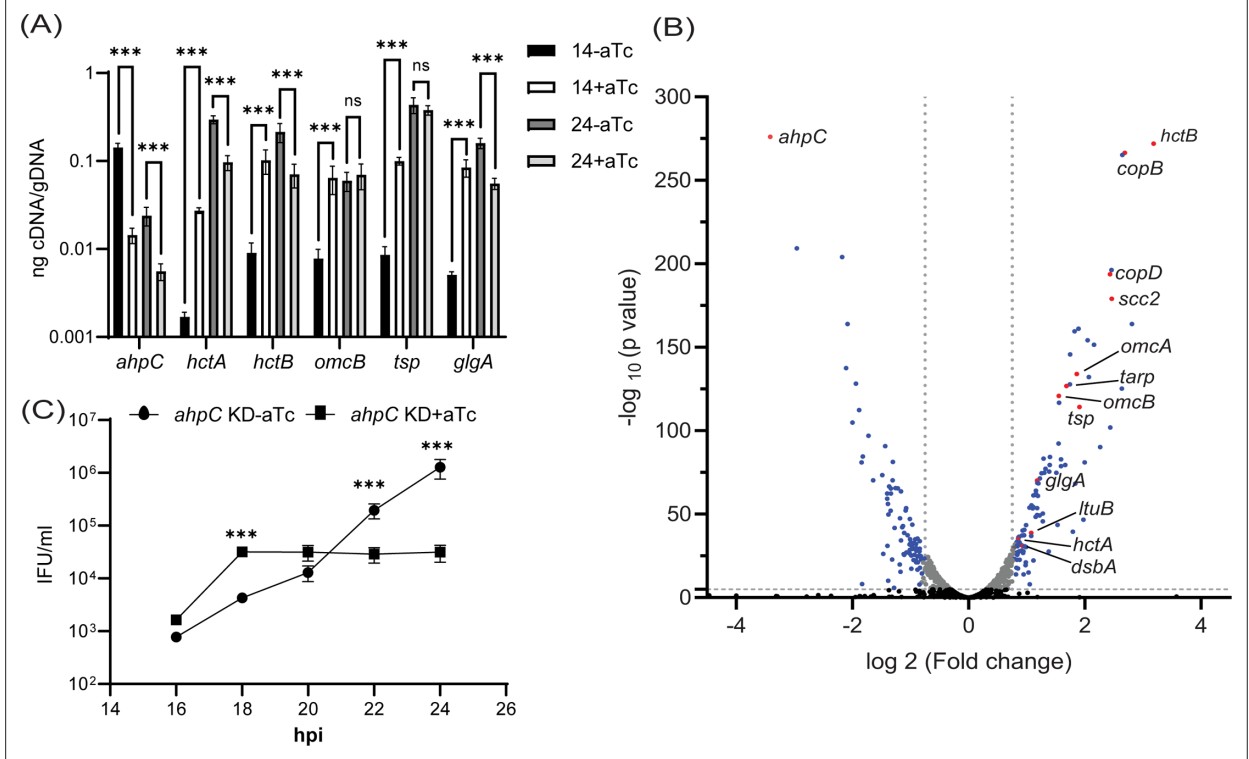

**Figure 8.** Effects of alkyl hydroperoxide reductase subunit C (*ahpC*) knockdown on chlamydial developmental cycle progression. (**A**) RT-qPCR analysis of late-cycle genes (*hctA*, *hctB*, *omcB*, *tsp*, and *glgA*) in *ahpC* knockdown. Experimental conditions were the same as mentioned in the legend of *Figure 4A*. Quantified cDNA was normalized to gDNA, and values were plotted on a log scale. ***$p<0.0001$ vs uninduced sample by using two-way ANOVA and Tukey's HSD was applied as post hoc test. Data represent three biological replicates. (**B**) Volcano plot of RNA-sequencing of *ahpC* knockdown. Experimental conditions were the same as mentioned in the legend of *Figure 4A*. The volcano plot was prepared using GraphPad Prism software. The vertical dashed lines indicate a fold change of 1.5 as compared to the respective uninduced control. The horizontal dashed line indicates a pvalue of 0.05. The black dots represent genes not significantly different and gray dots represent significantly altered but fold change lesser than 1.5. Blue and red spots represent statistically significant altered genes with more than a 1.5-fold change in transcription levels between uninduced and induced samples. Red dots indicate *ahpC* and canonical late genes also shown in *Table 1*. (**C**) One-step growth curve of *ahpC* knockdown. Samples were induced or not with 1 nM aTc at 10 hpi and harvested at 16, 18, 20, 22, and 24 hpi. IFUs recovered are displayed as log10 values. ***$p<0.0001$ vs uninduced sample by using two-way ANOVA and Tukey's HSD was applied as post hoc test. Data represent three biological replicates.

The online version of this article includes the following source data and figure supplement(s) for figure 8:

**Source data 1.** RT-qPCR (cDNA, gDNA), volcano plot of RNA-seq, and one step growth curve data of *ahpC* KD.

**Figure supplement 1.** Effects of alkyl hydroperoxide reductase subunit C (*ahpC*) knockdown/overexpression on chlamydial developmental cycle progression.

**Figure supplement 1—source data 1.** RT-qPCR (cDNA and gDNA) data of complemented and *ahpC* OE strains.

**Figure supplement 2.** Effect of penicillin treatment during alkyl hydroperoxide reductase subunit C (*ahpC*) knockdown.

**Figure supplement 2—source data 1.** RT-qPCR (cDNA and gDNA) data in the *ahpC* KD-spec strain.

measure IFUs at 2 hr intervals from 16 to 24 hpi. HeLa cells were infected, and knockdown was induced or not at 10 hpi with 1 nM aTc. At the indicated time points, IFU samples were collected and quantified. Consistent with our prediction, this experiment revealed higher IFUs (i.e. EBs) at 16 and 18 hpi during *ahpC* knockdown compared to the uninduced samples (*Figure 8C*). This difference was statistically significant at 18 hpi. However, further, EB production was stalled, with the uninduced strain continuing to produce EBs such that, by 24 hpi, there were significantly higher EB yields under these conditions (as noted in *Figure 4D*). These data show that the phenotypic consequence of higher expression at 14 hpi of genes functionally related to EBs is the concomitant earlier production of EBs. These data underscore that reduced activity of *ahpC* causes earlier secondary differentiation in *C. trachomatis*.

**Table 1.** List of canonical late genes significantly increased during *ahpC* knockdown at 14 hpi.

| CT # | CTL # | Name | Function | Fold change | Ref |
|------|-------|------|----------|-------------|-----|
| CT046 | CTL0302 | *hct2 (hctB)* | Histone H1-like protein HC2 | 9.10 | *Brickman and Hackstadt, 1993* |
| CT578 | CTL0841 | *copB* | Needle tip; translocator | 6.44 | *Ouellette et al., 2005* |
| CT576 | CTL0839 | *scc2 (lcrH_1)* | Type III secretion chaperone (Low calcium response protein H) | 5.52 | *Ouellette et al., 2005* |
| CT579 | CTL0842 | *copD* | Needle tip; translocator | 5.41 | *Ouellette et al., 2005* |
| CT441 | CTL0700 | *tsp* | Carboxy-terminal processing protease | 3.76 | *Swoboda et al., 2023* |
| CT444 | CTL0703 | *omcA* | Small cysteine-rich outer membrane protein | 3.63 | *Hatch, 1996* |
| CT456 | CTL0716 | *tarP* | Translocated actin-recruiting phosphoprotein | 3.22 | *Clifton et al., 2004* |
| CT443 | CTL0702 | *omcB* | Large cysteine-rich periplasmic protein | 2.94 | *Newhall, 1987* |
| CT798 | CTL0167 | *glgA* | Glycogen synthase | 2.26 | *Belland et al., 2003*; *Gehre et al., 2016* |
| CT080 | CTL0336 | *ltuB* | Late transcription unit B protein | 2.11 | *Belland et al., 2003* |
| CT177 | CTL0429 | *dsbA* | Disulfide bond chaperone | 1.91 | *Christensen et al., 2019* |
| CT743 | CTL0112 | *hctA* | Histone H1-like protein HC1 | 1.81 | *Barry et al., 1993* |

## *ahpC* knockdown activates transcription of late-cycle genes when bacterial replication is blocked

We detected earlier expression of EB-related genes during *ahpC* knockdown and were curious if the *ahpC* knockdown condition could activate these genes under conditions when the chlamydial developmental cycle is blocked. Pathogenic *Chlamydia* species undergo a polarized cell division process, in which peptidoglycan is transiently synthesized only at the division septum (*Abdelrahman et al., 2016*; *Cox et al., 2020*; *Ouellette et al., 2020*). As a result, during penicillin treatment, the division of RBs is blocked (*Moulder, 1993*; *Moulder et al., 1956*; *Ouellette et al., 2020*). However, the bacteria continue to grow in size resulting in aberrantly enlarged RBs (*Barbour et al., 1982*; *Matsumoto and Manire, 1970*; *Ouellette et al., 2012*), in which EB-related genes are not transcribed and production of EBs is inhibited (*Ouellette et al., 2006*; *Panzetta et al., 2018*).

To examine this, we generated an *ahpC* KD strain with spectinomycin resistance (*ahpC* KD-spec) and validated its phenotype as being the same as the penicillin-resistant *ahpC* KD strain used in our prior experiments. This new strain, *ahpC* KD-spec, was used to infect HeLa cells, and knockdown was induced or not at 10 hpi with 1 nM aTc. At the same time, samples were treated or not with 1 unit per mL of penicillin (Pen). IFA controls from these different conditions demonstrated the expected phenotypes (*Figure 8—figure supplement 2A*). RNA samples were harvested at 16 and 24 hpi and processed for RT-qPCR (*Figure 8—figure supplement 2B-D*). Pen treatment caused aberrantly enlarged RBs, irrespective of *ahpC* KD, whereas *ahpC* KD itself caused smaller inclusions (*Figure 8—figure supplement 2A*). Similarly, *ahpC* KD resulted in reduced *ahpC* transcripts in the induced conditions irrespective of the presence of Pen (*Figure 8—figure supplement 2B*). Next, we quantified transcript levels for the late genes *hctA* and *hctB* in these different conditions (*Figure 8—figure supplement 2C&D*). As expected, in the uninduced and untreated conditions, both transcripts had higher expression at 24 hpi, indicating their regular developmental expression during the late stage of the developmental cycle. Consistent with prior observations (*Ouellette et al., 2006*), levels of these genes in uninduced + Pen conditions were low. As we previously noted (*Figure 8A*), the transcripts of *hctA* and *hctB* are higher in the induced than uninduced samples at 16 hpi in the absence of Pen. Interestingly, in the induced + Pen condition, these genes had higher expression at 24 hpi than the uninduced + Pen control. At 16 hpi, their expression was either higher or similar between both samples (i.e., induced + Pen and uninduced + Pen). Collectively, these data support our observation that *ahpC* knockdown activates the transcription of late-cycle genes – even under conditions where developmental cycle progression is blocked.

## Discussion

Many bacteria use redox sensing as a mechanism to control gene expression and subsequent morphologic transitions. Typically, these organisms use a redox-sensitive transcription factor, such as OxyR to sense these changes in redox conditions and alter gene expression appropriately. For example, in *Pseudomonas aeruginosa*, OxyR is involved in the regulation of cellular metabolism in addition to oxidative stress defense (*Wei et al., 2012*). In *Caulobacter crescentus*, redox has been reported as a regulator of metabolism and cell cycle progression (*Hartl et al., 2020*; *Narayanan et al., 2015*). Thus, the ability of bacteria to assess oxidative stress (e.g., an elevated level of ROS) and to make physiological adjustments by altering gene expression patterns that favor their growth and development is critical for their survival.

ROS generation is an inevitable condition for pathogens growing in aerobic conditions, and resistance against oxidative stress is a key survival mechanism (*Fang, 2011*). Hence, pathogens have evolved detoxifying proteins, such as peroxiredoxins, to eliminate ROS (*Dip et al., 2014*; *Wang et al., 2004*; *Yang et al., 2002*). AhpC has been reported to have a crucial role in bacterial physiology, survival, and virulence by scavenging ROS and RNI (*Cosgrove et al., 2007*; *Kimura et al., 2012*; *Loprasert et al., 2003*; *Oh and Jeon, 2014*). In the process of detoxification of peroxides and peroxynitrite, AhpC is converted into an oxidized dimer, requiring alkyl hydroperoxide reductase AhpF or AhpD to regenerate its activity (*Koshkin et al., 2004*; *Poole and Ellis, 1996*; *Wong et al., 2017*). *C. trachomatis* encodes AhpC (Ct603/CTL0866) but lacks any annotated homologs of AhpF or AhpD. Therefore, it remains an open question how AhpC activity is regulated. Some studies have mentioned *ahpC* as an iron-responsive gene in *Chlamydia* (*Brinkworth et al., 2018*; *Pokorzynski et al., 2019*), but there is no detailed investigation to date about the role of this crucial antioxidant in the growth and development of *Chlamydia*. Our study is the first to characterize a function of AhpC in chlamydial biology.

An essential aspect of the scavenging activity of AhpC for many bacterial pathogens in which it has been studied is that it works best with endogenous (i.e., low) levels of $H_2O_2$. These bacteria, such as *Staphylococcus aureus* and *Yersinia pseudotuberculosis*, encode catalases, which have high Km for hydrogen peroxide that serve a predominant role in scavenging exogenous $H_2O_2$ (*Cosgrove et al., 2007*; *Wan et al., 2021*). Catalases can detoxify $H_2O_2$ at high levels (millimolar levels) and are crucial in responding to external $H_2O_2$ stress (*Mishra and Imlay, 2012*). The *ahpC* mutants in these bacteria become sensitive to organic peroxides but resistant to $H_2O_2$ due to higher catalase activity as a compensatory response to the lack of AhpC (*Antelmann et al., 1996*; *Mongkolsuk et al., 2000*; *Ochsner et al., 2000*). Furthermore, simultaneous mutations in both catalase and *ahpC* result in drastically enhanced sensitivity to all oxidizing agents tested (*Cosgrove et al., 2007*; *Ezraty et al., 2017*; *Seaver and Imlay, 2001*). *C. trachomatis* does not encode a catalase gene (*Boncompain et al., 2014*; *Rusconi and Greub, 2013*), and hypersensitivity of *ahpC* knockdown to both inorganic and organic peroxides indicates the absence of catalase and establishes that AhpC is the primary scavenger of ROS in *C. trachomatis*. Contrary to studies from other bacterial systems, we did not observe significant effects of PN with altered AhpC expression. One possible explanation may be our study's concentration (1 mM) of PN. Higher concentrations of PN could not be used due to toxic effects on the host cell (*Figure 3—figure supplement 1*). The other possibility may be the presence of some other unknown mechanism(s) to detoxify PN. For example, *Chlamydia* also encodes a superoxide dismutase (SOD) that may prevent the accumulation of the necessary precursors that are needed for PN production. We are currently investigating the function of the chlamydial SOD enzyme.

Resistance to different oxidants as a result of overexpression of *ahpC* (*Figure 3*, *Figure 3—figure supplement 3*) is consistent with studies in other bacteria (*Sherman et al., 1996*; *Zuo et al., 2014*), signifying that AhpC is the principal defense mechanism against oxidative stress in *Chlamydia*. During *ahpC* knockdown, the attenuated activity of AhpC severely affected the growth of *Chlamydia* (*Figure 4*). Notably, the observed morphology of inclusions during *ahpC* knockdown is strikingly similar to *Chlamydia* treated with a high concentration (1 mM) of oxidizing agents used (*Figure 4*, *Figure 3—figure supplement 2*), suggesting both conditions result in increased oxidative stress to the organism. This was further supported by the increased sensitivity of *ahpC* KD to sublethal concentrations of oxidants (*Figure 6*, *Figure 6—figure supplement 1*), consistent with previous data from other bacteria (*Cosgrove et al., 2007*; *Seaver and Imlay, 2001*; *Storz et al., 1989*; *Zhang et al., 2019*). In *ahpC* KD, the ROS level was dramatically higher (*Figure 5*, *Figure 5—figure supplement 1*),

as observed in other bacterial systems (*Zhang et al., 2019*). Further, the addition of ROS scavengers to the culture medium rescued the negative phenotypes associated with *ahpC* knockdown (*Figure 7G and H*). This strongly suggests that, in the absence of AhpC, higher amounts of reactive oxygen species accumulate in *Chlamydia* and that the negative impact on growth during *ahpC* knockdown is due to highly oxidized conditions in the organism. Moreover, for the complete restoration of inclusion size, multiple doses of scavengers were required throughout the experiment, further emphasizing that the internal chlamydial environment is increasingly oxidized and/or susceptible to oxidation in the absence of AhpC. These findings suggest that AhpC in *Chlamydia* is essential, having a functional role even more extensive than previously reported in other bacteria.

In further support of this observation, we noticed upregulated transcripts of late-cycle genes such as *hctA*, *hctB*, *glgA*, *omcA*, *omcB*, *dsbA*, and *tsp* at an earlier time (14 hpi) in the chlamydial developmental cycle as a result of reduced AhpC activity (*Figure 8*). Notably, in the developmental cycle of *C. trachomatis*, only RBs are present at 14 hpi, and secondary differentiation starts after 16 hpi (*Abdelrahman and Belland, 2005*). Expression of most of the late-cycle genes occurs after this time in the developmental cycle (*Belland et al., 2003*). These late-cycle genes are well characterized for their association with EBs. The *hctA* and *hctB* genes encode histone-like proteins responsible for chromosomal condensation during the differentiation of RBs into EBs. The *hctA* gene is among the first to be transcribed in the late stage (*Chiarelli et al., 2020*). Tsp is a periplasmic protease thought to be crucial in the degradation of RB-specific periplasmic proteins during secondary differentiation in *C. trachomatis* (*Swoboda et al., 2023*). OmcA, OmcB, and DsbA are responsible for the rigid cell wall and osmotic stability of the EBs (*Christensen et al., 2019*; *Hatch, 1996*; *Newhall, 1987*). GlgA is a key enzyme for glycogen accumulation in the inclusion lumen at late stages of the developmental cycle (*Gehre et al., 2016*). One crucial point to consider is the redox status of *Chlamydia*; 14 hpi is the time when reduced RBs predominate with virtually no oxidized EBs detectable. However, we established that *ahpC* knockdown shifts the redox status of the bacteria towards oxidation. Hence, the earlier and significantly higher detection of transcripts for these late-cycle genes indicates earlier secondary differentiation in the *ahpC* knockdown. Importantly, *ahpC* complementation during knockdown restored the regular developmental expression of these late-cycle genes, further supporting that this phenotype was due to the highly oxidized conditions created by reduced AhpC activity. The conclusion of these data is that AhpC levels, and presumably activity, have a direct effect on secondary differentiation.

Secondary differentiation (differentiation from RBs to EBs) is an essential step for chlamydial growth and survival, but there is a dearth of information regarding the mechanisms of its regulation. EBs and RBs have significantly different proteomic repertoires (*Saka et al., 2011*; *Skipp et al., 2005*; *Skipp et al., 2016*), and our group has identified critical functions for the cytoplasmic ClpXP and ClpCP and periplasmic Tsp proteases during secondary differentiation (*Jensen et al., 2025*; *Pan et al., 2023*; *Swoboda et al., 2023*; *Wood et al., 2022*).The *tsp* gene, in addition to another late-cycle gene, *hctB*, is transcriptionally regulated by sigma factor 28 ($\sigma^{28}$) (*Douglas and Hatch, 2000*; *Hatch and Ouellette, 2023*). Another sigma factor, $\sigma^{54}$ has been linked to the regulation of outer membrane components, type III secretion system components, and other genes typically expressed late in development (*Hatch and Ouellette, 2023*; *Soules et al., 2020b*). In addition to the two minor sigma factors, $\sigma^{28}$ and $\sigma^{54}$, *Chlamydia* encodes one major sigma factor, $\sigma^{66}$, which is post-translationally regulated by the relative levels of RsbV1 (antagonist) and RsbW (anti-sigma factor) (*Thompson et al., 2015*). This Rsb system senses ATP availability and has been proposed to regulate $\sigma^{66}$ (*Thompson et al., 2015*). A related study from *Soules et al., 2020a* showed that TCA intermediates act as ligands for RsbU, thereby linking the Rsb system to the TCA cycle and ATP synthesis. Collectively, these data show that post-translational mechanisms drive secondary differentiation in *Chlamydia*. However, no definitive 'switch' that triggers this step has been identified.

How might increasing oxidation regulate secondary differentiation? *Chlamydia* lacks an identifiable ortholog of OxyR so some other mechanism must be employed. We speculate that increasing ROS production from the metabolic activity of the RB will overcome the innate and stochastic activity of AhpC to detoxify ROS. This could either directly alter the activity of specific proteins or lead to the accumulation of damaged proteins that in turn trigger secondary differentiation. These possibilities are not mutually exclusive. We propose a simple model to explain this (*Figure 9*). As secondary differentiation is asynchronous and RBs divide through an asymmetric budding mechanism (*Ouellette*

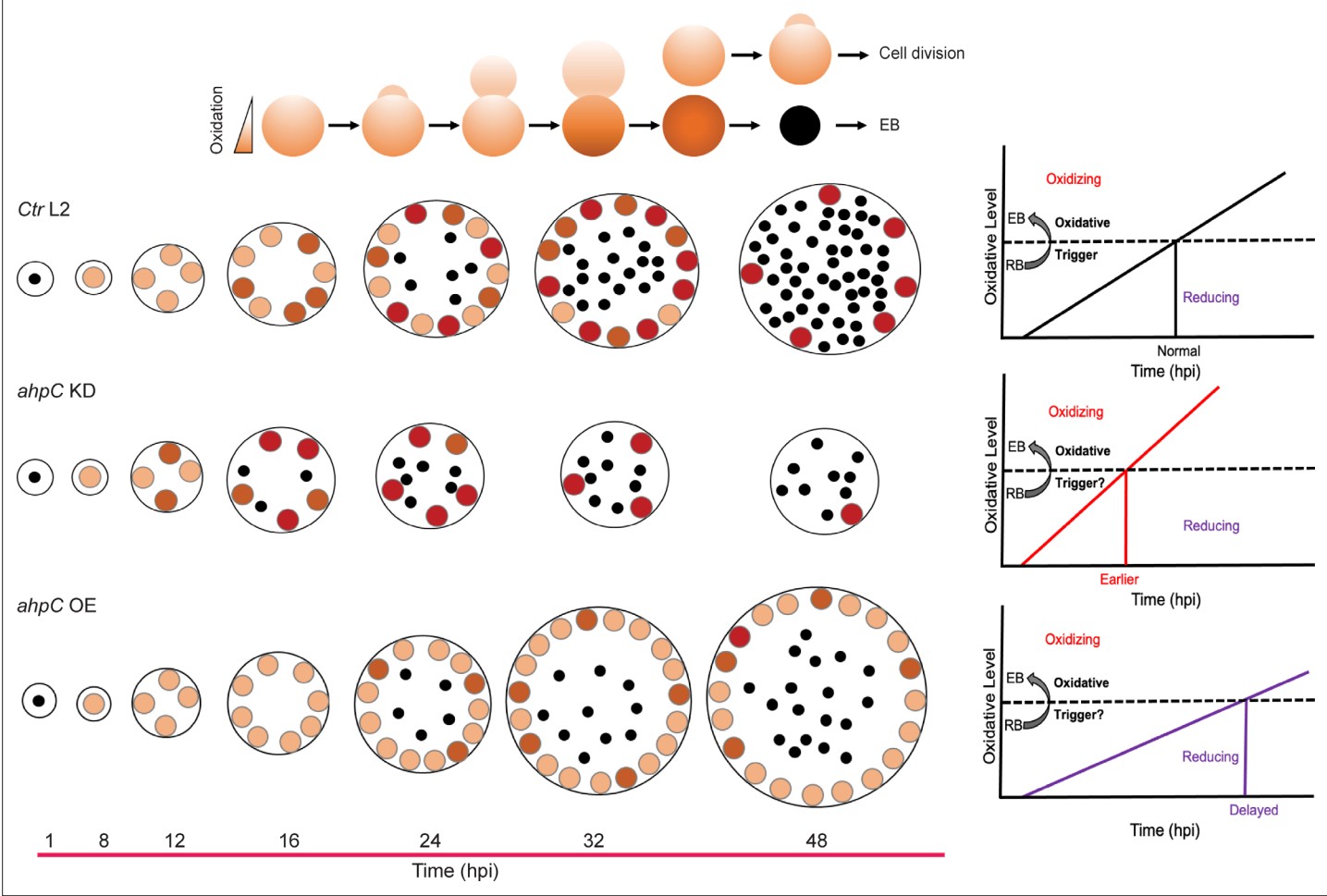

**Figure 9.** Altering the activity of alkyl hydroperoxide reductase subunit C (AhpC) in *Ctr* L2 impacts its developmental cycle progression. (Top) In *Chlamydia*, secondary differentiation is asynchronous and reticulate bodies (RBs) divide through an asymmetric budding mechanism. In such conditions, either the mother or daughter cell may inherit more oxidized proteins (represented by a darker shade), which can then impact whether a given RB will divide again or undergo secondary differentiation. (Bottom) The black dots represent EBs, the orange circles show RBs. The developmental cycle of wild-type *C. trachomatis* (*Ctr* L2) is shown. In *ahpC* KD, highly oxidized conditions lead some RBs to cross the oxidative threshold sooner, allowing activation of late genes and secondary differentiation earlier than other RBs. In *ahpC* overexpression, a reducing environment results in a delay in achieving the oxidative threshold, thus allowing RBs to continue to divide before committing to secondary differentiation.

*et al., 2020*), AhpC as well as proteins impacted by ROS levels are unevenly distributed between mother and daughter cell. This difference will lead some RBs to breach an oxidative threshold sooner, allowing activation of late genes and secondary differentiation earlier than other RBs. Consistent with this, the higher expression of genes functionally related to EBs and EB production during *ahpC* KD at an early stage of the chlamydial developmental cycle indicates earlier secondary differentiation as an outcome of diminished activity of *ahpC* in *C. trachomatis*. Consequently, increased late gene transcription should have the phenotypic effect of causing earlier production of EBs. Indeed, we quantified more IFUs in the *ahpC* knockdown at 16 and 18 hpi compared to the uninduced control condition. During *ahpC* KD, higher amounts of ROS accumulation in the bacterium create highly oxidized conditions. In contrast, *ahpC* overexpression scavenges ROS at a greater level leading to a more reducing environment. For the AhpC overexpression strain, the expression of late-cycle genes in the induced conditions compared to the uninduced control at 24 hpi is significantly lower with reduced production of EBs. These data support our hypothesis that the developmental cycle is delayed in these conditions with a concomitant delay in achieving the oxidative threshold, thus allowing RBs to continue to divide before committing to secondary differentiation. Taken together, our data directly link oxidation state and secondary differentiation in *Chlamydia*. Ongoing studies are focused on characterizing

redox-sensitive chlamydial proteins to understand which specific factors drive the shift from RBs to EBs (or EBs to RBs) in *C. trachomatis*.

# Materials and methods

## Strains and cell culture

For chlamydial transformation, McCoy mouse fibroblast cells (kind gift of Dr. Harlan Caldwell (NIH/NIAID)) were used. Human cervix adenocarcinoma epithelial HeLa cells (kind gift of Dr. Harlan Caldwell (NIH/NIAID)) were used for RT-qPCR, immunofluorescence assays (IFA), inclusion forming unit assays (IFU), viability assays, oxidative stress, RNA-seq, ROS measurements, and ROS scavenger assays. Both cell types were routinely grown and passaged in Dulbecco's modified Eagle's medium (DMEM; Gibco, #10-569-044) supplemented with 10% fetal bovine serum (FBS, Cytiva, #SH30396.03) and 10 µg/mL gentamicin (Gibco, #15710072) at 37°C and 5% CO2. All strains and cell types were verified to be Mycoplasma-negative using the LookOut mycoplasma PCR detection kit (Sigma, #MP0035-1KT). Cell lines were validated using ATCC STR-based services. For chlamydial transformations, *Chlamydia trachomatis* serovar L2 EBs lacking the endogenous pL2 plasmid (kind gift of Dr. Ian Clarke, University of Southampton) were used (*Wang et al., 2011*). Wild-type, density gradient-purified *Chlamydia trachomatis* 434/Bu (ATCC VR902B) EBs were used for sensitivity to oxidizing agents and ROS scavenger assays. Molecular biology reagents, oxidizing agents, CellROX Deep Red dye, and scavengers were purchased from Thermo Fisher unless otherwise noted.

## Plasmid construction

The primers, gBlock gene fragments, plasmids, and bacterial strains used for molecular cloning are listed in *Supplementary file 2* in the supplemental material. Constructs for chlamydial transformation were cloned using a high-fidelity (HiFi) cloning system from New England BioLabs (NEB, #E2621X). Primers were designed using the NEBuilder online primer generation tool (https://nebuilderv1.neb.com). For the overexpression strain, the *ahpC* gene was amplified by PCR with Phusion DNA polymerase (NEB, #M0530L) using *C. trachomatis* serovar L2 434/Bu genomic DNA as a template. The PCR product was purified using a PCR purification kit (Qiagen, #28506). The HiFi assembly reaction was performed as per the manufacturer's instructions in conjunction with the pBOMBDC plasmid digested with Fast Digest EagI and KpnI enzymes and dephosphorylated with FastAP (Thermo Fisher, #EF0652). The HiFi reaction mix was transformed into *E. coli* 10-beta (NEB, #C3019H). Plasmids were first confirmed by restriction enzyme digestion, and final verification of insert was performed using Sanger sequencing. A dCas12-based CRISPRi approach was used for *ahpC* knockdown generation (*Ouellette et al., 2021*). The pBOMBL12CRia plasmid encoding an anhydrotetracycline (aTc) inducible catalytically dead dCas12 protein was used as a vector. A crRNA targeting the 5' intergenic region of *ahpC* was designed and ordered as a presynthesized DNA fragment. For *ahpC* knockdown construct, 2 ng of the gBlock (Integrated DNA Technologies [IDT], Coralville, IA) listed in *Supplementary file 2* was combined with 25 ng of BamHI-digested, alkaline phosphatase-treated pBOMBL12CRia(e.v.)::L2 in a HiFi reaction according to the manufacturer's instructions (NEB). The plasmid was transformed into NEB 10-beta cells and verified by Sanger sequencing prior to transformation into *Chlamydia trachomatis*. To generate the complementation strain, the *ahpC* gene was amplified using primers listed in *Supplementary file 2* and fused with the *ahpC* knockdown construct (i.e. pBOMBL12CRia(*ahpC*)) digested with the Fast Digest restriction enzyme SalI (Thermo Fisher, #FD0644) and alkaline phosphatase-treated, using the HiFi reaction as mentioned above.

## Chlamydial transformation

Chlamydial transformations were performed using a protocol described previously, with some modifications (*Mueller et al., 2017*). One day before transformation, $1 \times 10^6$ McCoy cells were seeded in one six-well plate, and two wells were used per plasmid transformation. Briefly, for each well of the six-well plate, 2 µg of sequenced verified plasmids were incubated with $2.5 \times 10^6$ *C. trachomatis* serovar L2 without plasmid (-pL2) EBs in 50 µL Tris-CaCl$_2$ (10 mM Tris, 50 mM CaCl2, pH 7.4) at room temperature for 30 min. McCoy cells were washed with 2 mL Hank's Balanced Salt Solution (HBSS; Corning, #21–023-CV), and 1 mL HBSS was added back into each well. 1 mL of HBSS was added to each transformant mixture, and one well of a six-well plate was infected using this transformation solution.

Cells were centrifuged at 400×g for 15 min at room temperature followed by 15 min incubation at 37°C. HBSS was aspirated and replaced with antibiotic-free DMEM. At 8 hpi, 1 µg/mL of cycloheximide and 1 or 2 U/mL of penicillin G or 500 µg/mL spectinomycin were added to the culture media. The infection was passaged every 48 hr until a population of penicillin, or spectinomycin-resistant, green fluorescent protein (GFP) positive *C. trachomatis* was established. The chlamydial transformants were then serially diluted to isolate clonal populations. These isolated populations were further expanded and frozen at –80°C in a sucrose phosphate solution (2SP). To verify plasmid sequences, DNA was harvested from infected cultures using the DNeasy kit (Qiagen, #69506) and transformed into NEB 10-beta for plasmid propagation. Isolated plasmids were then verified by restriction digest and Sanger sequencing.

## Inclusion forming unit assay

Inclusion forming unit assay was performed to determine the infectious progeny (number of EBs) from a primary infection based on inclusions formed in a secondary infection. *C. trachomatis* transformants were infected into HeLa cells and induced or not with 1 nM aTc at 10 hpi. At 24 and 48 hpi, samples were harvested by scraping three wells of a 24-well plate in 2 sucrose-phosphate (2SP) solution and lysed via a single freeze-thaw cycle, serially diluted, and used to infect a fresh HeLa cell monolayer and allowed to grow for 24 hr. Samples were fixed with methanol, and stained with a goat antibody specific to *C. trachomatis* major outer membrane protein (MOMP; Meridian Biosciences, #B65266G) followed by staining with donkey anti-goat Alexa Fluor 594-conjugated secondary antibody (Invitrogen, #A-11058), and titers were enumerated by counting inclusions using a 20 x lens objective. All experiments were performed three times for three biological replicates. Induced values were expressed as a percentage of the uninduced values, which was considered as 100%.

## Immunofluorescence assay

HeLa cells were cultured on glass coverslips in 24-well tissue culture plates at $2×10^5$ cells/well, infected with the relevant strains, and, at 10 hpi, samples were induced or not with 1 nM aTc. At 24 hpi, samples were fixed and permeabilized using 100% methanol. Organisms were stained with anti-MOMP (Meridian Biosciences) primary antibody for all the strains and primary mouse anti-Cpf1 (dCas12) (Sigma-Millipore, #SAB4200756) for knockdown or complementation samples. Donkey anti-goat Alexa Fluor 594-conjugated secondary antibody (Invitrogen) was used to visualize *Chlamydia* in all the samples, and donkey anti-mouse Alexa Fluor 488-conjugated secondary antibody (Invitrogen, #A21202) was used for dCas12 expression. DAPI (Invitrogen, #D9542-5MG) was used for the visualization of host and bacterial cell DNA. These stained coverslips were mounted on glass slides using ProLong glass antifade mounting media (Invitrogen, #P36984) and imaged using a 100 x lens objective on a Zeiss Axioimager Z.2 equipped with Apotome.2 optical sectioning hardware and X-Cite Series 120PC illumination lamp using a 2MP Axiocam 506 monochrome camera.

## Nucleic acid extraction and RT-qPCR

HeLa cells were seeded in six-well tissue culture plates, infected with the *C. trachomatis* transformants, and induced or not at 10 hpi with 1 nM aTc. For each condition, triplicate wells were used for simultaneous harvest of RNA (for transcript analysis by RT-qPCR), gDNA (to normalize RT-qPCR data and quantification of gDNA), and IFA (to verify the morphological changes in samples in the tested conditions). For RNA extraction, cells were rinsed with DPBS twice and lysed with 1 mL TRIzol (Invitrogen, #15596018) per well as per the manufacturer's instructions. 200 µL of chloroform was added to extract the aqueous layer containing total RNA, which was precipitated with isopropanol. A total of 10 µg of purified RNA was treated with TURBO DNase (Invitrogen, #69506) according to the manufacturer's instructions to remove DNA contamination. DNA-free RNA was used for cDNA synthesis using random nonamers (N9; VWR, #101229–750) and SuperScript III reverse transcriptase (Invitrogen, #18-080-085) following the manufacturer's instructions. cDNA samples were diluted 10-fold with molecular biology-grade water and stored at –80°C. A total of 2.5 µL of each diluted cDNA sample was used per well of a 96-well qPCR plate. For each of the three biological replicates, each sample was analyzed in triplicate on a QuantStudio 3 system (Applied Biosystems) using the standard amplification cycle with melting curve analysis. For gDNA, one well of a six-well plate per condition was scraped in 500 µL DPBS, split in half (i.e. 250 µL), and frozen at –80°C. Each sample was then thawed and frozen

twice more for a total of three freeze/thaw cycles, and gDNA was extracted using the DNeasy DNA extraction kit (Qiagen, #69506) according to the manufacturer's guidelines. The isolated gDNA was quantified and diluted down to 5 ng/µL prior to use in quantitative PCR (qPCR). A total of 2.5 µL of each diluted gDNA sample was mixed with 10 µL of PowerUp SYBR green master mix (Applied Biosystems, #A25778) in a 96-well qPCR plate and was analyzed on a QuantStudio 3 system (Applied Biosystems). Each sample from each biological replicate was tested in triplicate. For each primer set used, a standard curve of gDNA was generated against purified *C. trachomatis* L2 genomic DNA, and the cDNA levels were normalized to gDNA levels or 16S rRNA for analysis. All experiments were performed three times for three biological replicates. At the same time, morphological differences were monitored in the IFA control, with samples fixed with methanol at the time of harvesting of RNA/gDNA. Staining and imaging were performed as described above.

## Enrichment, library preparation, and statistical analyses of RNA-sequencing samples

Three biological replicates were prepared and processed for RNA isolation as mentioned above. Enrichment of microbial RNA was performed using MICROB*Enrich* (Invitrogen, #AM1901) and MICROB*Express* (Invitrogen, #AM1905) kits as per the manufacturer's instructions. Samples were aliquoted, stored at –80°C, and submitted to the UNMC Genomics Core. Before library preparation, quality control (QC) was assessed using Nanodrop and Fragment Analyzer (Advanced Analytical). Final libraries were quantified using Qubit DS DNA HS Assay reagents in Qubit Fluro meter (Life Technologies), and the size of the libraries was measured via Fragment Analyzer. Individual RNA-seq libraries were prepared using 400 ng of total RNA by Illumina Ribo-Zero plus Microbiome (Illumina, Inc San Diego, CA) and were multiplexed and subjected to 100 bp paired read sequencing to generate >20 million pairs of reads per sample using Midoutput NextSeq 300 cycle kit by the UNMC Genomics Core facility (*Supplementary file 1*). The original fastq format reads was trimmed by the fqtrim tool (https://ccb.jhu.edu/software/fqtrim) to remove adapters, terminal unknown bases (Ns), and low-quality 3' regions (Phred score <30). The trimmed fastq files were processed by FastQC (*Andrews, 2010*). *Chlamydia trachomatis* 434/Bu bacterial reference genome and annotation files were downloaded from Ensembl (here). Sequencing data were analyzed by the Bioinformatics and Systems Biology Core (BSBC). The false discovery rate (FDR) and Bonferroni adjusted p values were provided to adjust for multiple-testing problem. Fold changes were calculated from the GLM, which corrects for differences in library size between the samples and the effects of confounding factors. The trimmed fastq files were mapped to *Chlamydia trachomatis* 434/Bu by CLC Genomics Workbench 23 for RNA-seq analyses.

## Viability assay

Cell viability assays were performed using PrestoBlue (Invitrogen, #A13261). HeLa cells were seeded in 96 well plates and were infected or not with the indicated chlamydial strains (e.g. pBOMBDC.ev). Samples were treated or not as per the experimental parameters (e.g., different concentrations of oxidizing agents or scavengers). Medium without cells was used as the blank control. At the end point, 10% PrestoBlue (v/v) was added in the wells and incubated at 37°C for 30 min while protecting the plate from light. Fluorescence was read at excitation 560 nm and emission 590 nm using an Infinite M200 Pro plate reader (Tecan). The treated values were expressed as a percentage of the untreated values, which was considered as 100%. All experiments were performed three times for three biological replicates.

## Oxidizing agents' susceptibility testing

Susceptibility of host cells and chlamydiae to different oxidizing agents was tested in HeLa cells infected with density gradient-purified *Chlamydia trachomatis* 434/Bu EBs or *C. trachomatis* transformants. For *C. trachomatis* transformants, expression of the construct was induced or not with 1 nM aTc at 10 hpi. At 16 hpi, different concentrations of oxidizing agents were added, and samples were incubated at 37°C for 30 min. These samples were washed three times with HBSS, fresh DMEM media was added to the wells, and cultures were allowed to grow until 24 hpi. At 24 hpi, inclusion forming unit assays and immunofluorescence analysis were performed as described above.

## ROS scavenging by chemical compounds

The scavenging capacity of different ROS scavengers (DMTU, Sigma-Aldrich, #D188700 and α-Tocopherol, Sigma-Aldrich, #T3251-5G) was tested using HeLa cells infected with *Chlamydia trachomatis* 434/Bu EBs or the *ahpC* KD strain. To examine the protective effect of scavengers in the absence or presence of oxidative stress, *Chlamydia trachomatis* 434/Bu EBs infected HeLa cells were used. In these samples, scavengers were added or not at 10 hpi in respective wells. At 16 hpi, wells were washed three times with HBSS, and fresh DMEM containing 500 µM $H_2O_2$ or not was added. After 30 min, wells were washed three times with HBSS, and fresh DMEM containing scavengers or not was added back in the respective wells, which were incubated until 24 hpi prior to collecting samples for IFU assay and IFA analysis. The effect of these scavengers was further tested in *ahpC* KD. HeLa cells were infected with *ahpC* KD, scavengers were added or not at 9.5 hpi before induction at 10 hpi with 1 nM aTc. At 14 hpi, scavengers were added again in the respective wells, and, at 24 hpi, samples were collected for IFU assay and IFA.

## ROS detection by CellROX Deep Red

Intracellular ROS levels were measured in uninfected and *ahpC* KD-infected HeLa cells using CellROX Deep Red dye (Invitrogen, #C10422). Samples were induced or not at 10 hpi with 1 nM aTc. Medium without cells was used as the blank control. At the end point, media was removed, and samples were washed three times with DPBS and incubated with CellROX Deep Red dye for 30 min in the dark at 37°C. Fluorescence was read at excitation 644 nm and emission 665 nm using an Infinite M200 Pro (Tecan). Microscopic images were captured using live cells at 24 or 40 hpi. Experimental conditions were the same as mentioned above. All experiments were performed three times for three biological replicates.

## Materials availability

Any unique resources developed in this study will be made available upon request.

## Acknowledgements

We thank Dr. H Caldwell (NIH/NIAID) for providing eukaryotic cell lines and Dr. I Clarke (University of Southampton) for providing the plasmidless strain of *C trachomatis* serovar L2. We thank Dr. Elizabeth A Rucks for critical feedback. We thank the members of the Rucks/Ouellette research group for the thoughtful discussion of the material presented. Funding for this work was provided by the National Institutes of Health (NIH/NIAID) grants R01AI170688 and R21AI178150 to SPO. Authors thank the UNMC Genomics Core Facility and Bioinformatics and Systems Biology Core (BSBC) at UNMC for RNA-seq services. The UNMC Genomics Core Facility receives partial support from the National Institute for General Medical Science (NIGMS) INBRE - P20GM103427-19, as well as the National Cancer Institute (NCI) and The Fred & Pamela Buffett Cancer Center Support Grant- P30CA036727. The BSBC receives support from the Nebraska Research Initiative (NRI) and NIH (2P20GM103427, 5P30CA036727, and 2U54GM115458).

## Additional information

### Funding

| Funder | Grant reference number | Author |
| --- | --- | --- |
| National Institutes of Health | R01AI170688 | Scot P Ouellette |
| National Institutes of Health | R21AI178150 | Scot P Ouellette |

The funders had no role in study design, data collection and interpretation, or the decision to submit the work for publication.

## Author contributions
Vandana Singh, Conceptualization, Formal analysis, Validation, Investigation, Visualization, Methodology, Writing - original draft; Scot P Ouellette, Conceptualization, Formal analysis, Supervision, Funding acquisition, Methodology, Project administration, Writing – review and editing

## Author ORCIDs
Vandana Singh ⬤ https://orcid.org/0000-0001-5368-6200
Scot P Ouellette ⬤ https://orcid.org/0000-0002-3721-6839

Reviewer #1 (Public review): https://doi.org/10.7554/eLife.98409.3.sa1
Reviewer #2 (Public review): https://doi.org/10.7554/eLife.98409.3.sa2
Author response https://doi.org/10.7554/eLife.98409.3.sa3

## Additional files

### Supplementary files
Supplementary file 1. Summary of alkyl hydroperoxide reductase subunit C (*ahpC*) knockdown RNA-seq results.

Supplementary file 2. List of plasmids, strains, and primers used in this study.

MDAR checklist

### Data availability
The raw and processed RNA sequencing reads in fastq format have been deposited in Gene Expression Omnibus (GEO) under Accession number GSE278846. All other data generated or analysed during this study are included in the manuscript and supporting files; source data files have been provided where appropriate.

The following dataset was generated:

| Author(s) | Year | Dataset title | Dataset URL | Database and Identifier |
|---|---|---|---|---|
| Ouellette et al. | 2024 | Altering the redox status of *Chlamydia trachomatis* directly impacts its developmental cycle progression | https://www.ncbi.nlm.nih.gov/geo/GSE278846 | NCBI Gene Expression Omnibus, GSE278846 |

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
