## [Editor Report · eLife Assessment]

In this **valuable** study, the authors propose a model wherein the bacterial redox state plays a crucial role in the differentiation of *Chlamydia trachomatis* into elementary and reticulate bodies. They provide **solid** evidence to argue that a highly oxidising environment favours the formation of elementary bodies while a reducing condition slows down development. Overall, the study **convincingly** demonstrates that Chlamydial redox states play a role in differentiation, an observation that may have implications for the study of other bacterial systems.

---

## [Referee Report · Reviewer #1 (Public review)]

Summary:

Chlamydia spp. has a biphasic developmental cycle consisting of an extracellular, infectious form called an elementary body (EB) and an intracellular, replicative form known as a reticular body (RB). The structural stability of EBs is maintained by extensive cross linking of outer membrane proteins while the outer membrane proteins of RBs are in a reduced state. The overall redox state of EBs is more oxidized than RBs. The authors propose that redox state may be a controlling factor in the developmental cycle. To test this, alkyl hydroperoxide reductase subunit C (ahpC) was overexpressed or knocked down to examine effects on developmental gene expression. KD of ahpC induced increased expression of EB-specific genes and accelerated EB production. Conversely, overexpression of phpC delayed differentiation to EBs. The results suggest that chlamydial redox state may play a role in differentiation.

Strengths:

Uses modern genetic tools to explore the difficult area of temporal gene expression throughout the chlamydial developmental cycle.

Weaknesses:

The environmental signals triggering ahpC expression/activity are not determined.

Comments on revisions:

I am satisfied with the modifications made to the manuscript.

---

## [Referee Report · Reviewer #2 (Public review)]

The factors that influence the differentiation of EBs and RBs during Chlamydial development are not clearly understood. A previous study had shown a redox oscillation during the Chlamydial developmental cycle. Based on this observation, the authors hypothesize that the bacterial redox state may play a role in regulating the differentiation in Chlamydia. To test their hypothesis, they make knock-down and overexpression strains of the major ROS regulator, ahpC. They show that the knock-down of ahpC leads to a significant increase in ROS levels leading to an increase in the production of elementary bodies and overexpression leads to a decrease in EB production likely caused by a decrease in oxidation. From their observations, they present an interesting model wherein an increase in oxidation favors the production of EBs.

Comments on revisions:

Major concerns have been satisfactorily addressed.

---

## [Author Response]

The following is the authors’ response to the original reviews.

**Public Reviews:**

**Reviewer #1 (Public Review):**
Summary:Chlamydia spp. has a biphasic developmental cycle consisting of an extracellular, infectious form called an elementary body (EB) and an intracellular, replicative form known as a reticular body (RB). The structural stability of EBs is maintained by extensive cross-linking of outer membrane proteins while the outer membrane proteins of RBs are in a reduced state. The overall redox state of EBs is more oxidized than RBs. The authors propose that the redox state may be a controlling factor in the developmental cycle. To test this, alkyl hydroperoxide reductase subunit C (ahpC) was overexpressed or knocked down to examine effects on developmental gene expression. KD of ahpC induced increased expression of EB-specific genes and accelerated EB production. Conversely, overexpression of ahpC delayed differentiation to EBs. The results suggest that chlamydial redox state may play a role in differentiation.Strengths:Uses modern genetic tools to explore the difficult area of temporal gene expression throughout the chlamydial developmental cycle.Weaknesses:

The environmental signals triggering *ahpC* expression/activity are not determined.

Thank you for your comments. Our data and those of others have shown that *ahpC* is expressed as a mid-developmental cycle gene (i.e., when RBs predominate in the population). This is true of most chlamydial genes, and the factors that determine developmental expression are not fully understood. As we noted in the Discussion, *Chlamydia* lacks AhpF/D orthologs, so it is not clear how AhpC activity is regulated. Related to determining environmental signals that trigger activity of AhpC, then this is a non-trivial issue in an obligate intracellular bacterium. Our assumption is that AhpC is constitutively active and that the increasing metabolic production of ROS eventually overcomes the innate (and stochastic) activity of AhpC to handle it, hence the threshold hypothesis. Importantly, the stochasticity is consistent with what we know about secondary differentiation in *Chlamydia*. We have tried to clarify these points in the Discussion.

**Reviewer #2 (Public Review):**
The factors that influence the differentiation of EBs and RBs during Chlamydial development are not clearly understood. A previous study had shown a redox oscillation during the Chlamydial developmental cycle. Based on this observation, the authors hypothesize that the bacterial redox state may play a role in regulating the differentiation in Chlamydia. To test their hypothesis, they make knock-down and overexpression strains of the major ROS regulator, ahpC. They show that the knock-down of ahpC leads to a significant increase in ROS levels leading to an increase in the production of elementary bodies and overexpression leads to a decrease in EB production likely caused by a decrease in oxidation. From their observations, they present an interesting model wherein an increase in oxidation favors the production of EBs.Major concern:In the absence of proper redox potential measurements, it is not clear if what they observe is a general oxidative stress response, especially when the knock-down of *ahpC* leads to a significant increase in ROS levels. Direct redox potential measurement in the *ahpC* overexpression and knock-down cells is required to support the model. This can be done using the roGFP-based measurements mentioned in the Wang et al. 2014 study cited by the authors.

Thank you for this suggestion. It is definitely something that we are looking to implement. However, our current vectors don’t allow for roGFP2 in combination with inducible expression of a gene of interest. We will need to completely redesign our vectors, and, in *Chlamydia*, the validation of such new vectors together with *ahpC* OE or KD may literally take a year or longer.

In lieu of this, we used the CellRox redox reactive dye to image live chlamydiae during normal growth or *ahpC* KD. During *ahpC* KD, these organisms are clearly much brighter than the control, uninduced conditions. These data are included as new Figure 5 to go along with the data we previously reported from the plate reader measurements. These data also clearly indicate that the readings we observed are from *Chlamydia* and not the host cell.

As far as a general oxidative stress response, *Chlamydia* lacks any transcriptional regulators akin to OxyR. The response we’ve measured, earlier expression of genes associated with secondary differentiation, would be an odd stress response not consistent with a focused program to respond to oxidative stress. We added new RNAseq data further showing this effect of a global earlier increase in late gene transcripts.

**Reviewer #3 (Public Review):**
Summary:The study reports clearly on the role of the AhpC protein as an antioxidant factor in *Chlamydia trachomatis* and speculates on the role of AhpC as an indirect regulator of developmental transcription induced by redox stress in this differentiating obligate intracellular bacterium.Strengths:The question posed and the concluding model about redox-dependent differentiation in chlamydia is interesting and highly relevant. This work fits with other propositions in which redox changes have been reported during bacterial developmental cycles, potentially as triggers, but have not been cited (examples PMID: 2865432, PMID: 32090198, PMID: 26063575). Here, AhpC over-expression is shown to protect Chlamydia towards redox stress imposed by H2O2, CHP, TBHP, and PN, while CRISPRi-mediated depletion of AhpC curbed intracellular replication and resulted in increased ROS levels and sensitivity to oxidizing agents. Importantly, the addition of ROS scavengers mitigated the growth defect caused by AhpC depletion. These results clearly establish the role of AhpC affects the redox state and growth in Ct (with the complicated KO genetics and complementation that are very nicely done).Weaknesses:However, with respect to the most important implication and claims of this work, the role of redox in controlling the chlamydial developmental cycle rather than simply being a correlation/passenger effect, I am less convinced about the impact of this work. First, the study is largely observational and does not resolve how this redox control of the cell cycle could be achieved, whereas in the case of _Caulobacte_r, a clear molecular link between DNA replication and redox has been proposed. How would progressive oxidation in RBs eventually trigger the secondary developmental genes to induce EB differentiation? Is there an OxyR homolog that could elicit this change and why would the oxidation stress in RBs gradually accumulate during growth despite the presence of AhpC? In other words, the role of AhpC is simply to delay or dampen the redox stress response until the trigger kicks in, again, what is the trigger? Is this caused by increasing oxidative respiration of RBs in the inclusion? But what determines the redox threshold?

Thank you for your comments. As the reviewer notes, our work clearly demonstrates that AhpC acts as an antioxidant in *Chlamydia trachomatis.* Further, we have shown that transcription of the late cycle genes is altered upon altered activity of AhpC, which acts as a proof of concept that redox is (one of) the key factor(s) controlling developmental cycle progression in *Chlamydia*. Our new RNAseq data indicate that a broad swath of well characterized late genes is activated, which contradicts the argument that what we’ve measured is a stress response (unless activation of late genes in *Chlamydia* is generally a stress response (not the case based on other models of stress) – in which case we would not be able to differentiate between these effects). We hypothesize that ROS production from the metabolic activities of RBs serves as a signal to trigger secondary differentiation from RBs to EBs. How this exact threshold is determined is currently unknown as *Chlamydia* does not have any annotated homolog for OxyR. It is beyond the scope of the present manuscript to identify and then characterize what specific factor(s) control(s) this response. We fully anticipate that multiple factors are likely impacted by increasing oxidation, so dissecting the exact contributions of any one factor will represent (a) potential separate manuscript(s). Nonetheless, this remains an overarching goal of the lab yet remains challenging given the obligate intracellular nature of *Chlamydia*. Strategies that would work in a model system, like *Caulobacter*, that can be grown in axenic media are not easily implemented in *Chlamydia*.

As we noted above in another response, *ahpC* is transcribed as a mid-cycle gene with a peak of transcription corresponding to the RB phase of growth. We hypothesize that the gradual accumulation of ROS from metabolic activity will eventually supercede the ability of AhpC to detoxify it. This would result in any given RB asynchronously and stochastically passing this threshold (and triggering EB formation), which is consistent with what we know about secondary differentiation in *Chlamydia*.

I also find the experiment with Pen treatment to have little predictive power. The fact that transcription just proceeds when division is blocked is not unprecedented. This also happens during the Caulobacter cell cycle when FtsZ is depleted for most developmental genes, except for those that are activated upon completion of the asymmetric cell division and that is dependent on the completion of compartmentalization. This is a smaller subset of developmental genes in caulobacter, but if there is a similar subset that depends on division on chlamydia and if these are affected by redox as well, then the argument about the interplay between developmental transcription and redox becomes much stronger and the link more intriguing. Another possibility to strengthen the study is to show that redox-regulated genes are under the direct control of chlamydial developmental regulators such as Euo, HctA, or others and at least show dual regulation by these inputs -perhaps the feed occurs through the same path.

Comparisons to other model systems are generally of limited value with *Chlamydia*. All chlamydial cell division genes are mid-cycle (RB-specific) genes, just like *ahpC*. There is no evidence of a redox-responsive transcription factor (whether EUO, HctA, or another) that activates or represses a subset of genes in *Chlamydia*. Similarly, there is no evidence that redox directly and specifically impacts transcription of cell division genes based on our new RNAseq data. The types of experiments suggested are not easily implemented in *Chlamydia*, but we would certainly like to be able to do them.

As it pertains to penicillin specifically, we and others have shown that treating chlamydiae with Pen blocks secondary differentiation (meaning late genes are not transcribed). Effectively, Pen treatment freezes the organism in an RB state with continued transcription of RB genes. What we have shown is that, even during Pen treatment (which blocks late gene transcription), *ahpC* KD can overcome this block, which shows that elevated oxidation is able to trigger late gene expression even when the organisms are phenotypically blocked from progressing to EBs. The comparison from our perspective to *Caulobacter* is of limited value.

This redox-transcription shortcoming is also reflected in the discussion where most are about the effects and molecular mitigation of redox stress in various systems, but there is little discussion on its link with developmental transcription in bacteria in general and chlamydia.

We have edited the Discussion to include a broader description of the results and included additional citations as suggested by the reviewer (PMID: 32090198, PMID: 26063575). However, we found one suggested article (PMID: 2865432) is not relevant to our study, so we didn’t cite it in our present manuscript. There may have been a typo, so feel free to provide us the correct PMID that can be cited.

**Reviewer #1 (Recommendations For The Authors):**
(1) Line 146. A minor point, but inclusion-forming units directly measure infectious EBs. In some cases, the particle-to-infectivity ratio approaches unity. I don't believe IFUs are a "proxy".

Following reviewers comment we have modified the statement.

(2) Figure 2E. Results are normalized to uninduced. The actual number of IFUs in the uninduced should be provided.

In the revised version of the manuscript, we have provided actual number of IFUs at 24 and 48 hpi in the uninduced condition of both *ahpC* OE and EV.

(3) Figures 3B&D. The shades of gray are not possible to distinguish. I'd suggest color or direct labeling.

Following reviewer’s suggestion, in the latest version of the manuscript we have replaced gray shaded graphs with RGB colored graphs for better visualization and understanding.

(4) Lines 217-224, Figure 4. Is it possible to get micrographs of the reporter retention in chlamydiae to demonstrate that it is chlamydial ROS levels that are being measured and not cellular?

Following reviewer’s comment, we performed live-cell microscopy using uninfected HeLa cells and *ahpC* KD in the uninduced and induced conditions at 24 and 40 hpi. We have created new Fig. 5A with the quantitative ROS measurement graph done by the plate reader (old figure 4 E) and these new 24 hpi/40 hpi microscopy images (Fig 5B and S4).

(5) The Discussion is overly long and redundant. Large portions of the discussion are simply a rehash of the Results listing by figure number the relevant conclusions.

Following reviewer’s suggestion, the discussion is modified.

**Reviewer #2 (Recommendations For The Authors):**
(1) In Figure 2, *ahpC* is significantly overexpressed at 14 hpi. An IFA as in 2B for 14hpi will be useful. This will help to understand how quick the effect of ahpC overexpression is on development.

We have added 14 hpi IFA of *ahpC* and EV as part of Fig 2B.

(2) In Figure 2E, is there a reason that there is no increase in recoverable IFUs between 24h and 48h for the EV?

The graph in 2E is % of uninduced. For more clarity, we have mentioned absolute IFUs of uninduced samples in the revised manuscript and IFU level at 48 hpi IFU is higher than the 24 hpi.

(3) In Figure 3, Can relative levels of RB vs EB measured? This will provide a convincing case for the production of more EBs even when only less/more RBs are present. The same stands for Figure 4.

We assumed that the comment is for Fig. 2 not the Fig. 3 and following reviewer’s constructive suggestion, we have attempted to resolve the issue. We normalized log10 IFUs/ml with log10 gDNA for 24 hpi and added as figure 2F and 4E. This may resolve the reviewer’s concern about the levels of RBs and EBs.

(4) A colour-coded Figure 3B and D, instead of various shades of grey, will be easy for the reader to interpret.

Agreed with the reviewer. For better visualization and understanding of the data, we have replaced gray shaded graphs with RGB colored graphs in the latest version of the manuscript.

**Reviewer #3 (Recommendations For The Authors):**
Other comments:(1) The first paragraph of the discussion should be deleted. It's not very useful or revealing and just delivers self-citations.

Following reviewer’s suggestion, we rewrote the discussion.

(2) The first paragraph of the results section does not present results. It's an intro.

We incorporated this information into the Intro as suggested.

(3) Has the redox difference between RBs and EBs been experimentally verified by these authors as depicted and claimed in Figure 1A with the cell-permeable, fluorogenic dye CellROX Deep Red for example? It is important to confirm this for EBs and RBs in this setup.

The difference between redox status of RBs and EBs is studied and established before by previous studies such as Wang et al., 2014.

(4) l77. Obligate intracellular alpha-proteobacteria also differentiate ... not only chlamydiae.

We have modified the sentence.

(5) l127. Is the redox state altered upon *ahpC* overexpression?

The *ahpC* overexpression strain showed hyper resistance for the tested oxidizing agents (including the highest concentration tested) indicating highly reduced conditions as a result of higher activity of AhpC.